# Radiative forcing and stratospheric ozone changes due to major forest fires and recent volcanic eruptions including Hunga Tonga

Christoph Brühl[1], Matthias Kohl[1], and Jos Lelieveld[1]

[1]Max Planck Institute for Chemistry, Mainz, Germany

**Correspondence:** christoph.bruehl@mpic.de

**Abstract.** Recent studies suggest that emissions from large forest fires affect stratospheric chemistry, dynamics, and climate, similar to major volcanic eruptions. Using the chemistry-climate model EMAC, we demonstrate that organic carbon emitted from forest fires, injected into the stratosphere through pyro-cumulonimbi, enhances heterogeneous chlorine activation related to high solubility of HCl in particles containing organic acids and an augmented aerosol surface area, in agreement with existing literature. Following the 2019/2020 Australian megafires, the upward transport of the pollution plumes resulted in enhanced ozone depletion in the Southern Hemisphere lower stratosphere, as corroborated by satellite observations. It diminished column ozone in the following two years, accompanied by a dynamically induced reduction in 2020 due to the lofting of smoke-filled vortices, in total by up to 45 DU. The eruption of the submarine Hunga Tonga volcano in January 2022 caused a decline in total ozone across the entire Southern Hemisphere. The water vapour injection from the volcano altered only the vertical distribution of ozone loss. The sunlight-absorbing aerosol from the Australian and, to a smaller extent, the Canadian forest fire emissions in 2019/2020 induced the most significant perturbation in stratospheric optical depth since the major eruption of Pinatubo in 1991. It shifted the sign of instantaneous stratospheric aerosol forcing, derived at the top of the atmosphere, from -0.2 $\mathrm{Wm^{-2}}$ to +0.3 $\mathrm{Wm^{-2}}$ in January 2020. The global aerosol radiative forcing resulting from the Hunga Tonga eruption was -0.16 $\mathrm{Wm^{-2}}$, primarily driven by changes in stratospheric sulfate aerosols. The positive radiative forcing from the injected water vapour was minimal.

## 1  Introduction

About 20 years ago Fromm et al. (2004, 2005) realized first, that major Boreal forest fires are able to inject material into the stratosphere with global consequences for dynamics and ozone. Since 2017 it was systematically observed that these fires which inject smoke particles via pyro-cumulonimbi directly into the midlatitude stratosphere can have a global impact on the radiation budget (Yu et al., 2021; Senf et al., 2023), stratospheric dynamics (Yu et al., 2019; Damany-Pearce et al., 2022; Khaykin et al., 2020; Allen et al., 2020; Kablick et al., 2020; Santee et al., 2022), and heterogeneous chemistry of the stratosphere (Solomon et al., 2022, 2023; Stone et al., 2025; Ohneiser et al., 2022; Strahan et al., 2022; Salawitch and McBride, 2022; Ansmann et al., 2022). This holds for the Australian fires in December 2019 to January 2020 as well as Canadian fires in 2017 (Kloss et al., 2019; Doglioni et al., 2022) and 2019. There are also the Canadian fires in 2023. Vescovini et al. (2024) address the Californian fires in 2020 and the Siberian fires in 2019. The latter 3 events are not considered in our study.

Because of the absorbing aerosol the radiative impact of forest fire plumes differs from the one of major volcanic eruptions like for example Hunga Tonga in January 2022 (Taha et al., 2022) or Raikoke in 2019 where scattering aerosol dominates. Radiative heating of the absorbing smoke particles causes self-lofting and in case of the Australian fires synoptic scale anticyclonic vortices lifting lower stratospheric and upper tropospheric smoke containing airmasses with high $N_2O$ and low ozone to the middle stratosphere (Allen et al., 2020; Kablick et al., 2020; Ma et al., 2024). Some of these vortices acting like a containment vessel transport the smoke particles to high latitudes. This and the about 10 to 15 km thick layer in the lower stratosphere perturbed by smoke (Khaykin et al., 2020) cause a deeper Antarctic ozone hole due to heterogeneous chlorine activation (Stone et al., 2021, 2025; Cutts, 2023).

Hunga Tonga (now shortly referred to as Hunga (Santee et al., 2024)), the most explosive eruption in the last 3 decades, has been analyzed in several studies (Khaykin et al., 2022; Li et al., 2024; Wilmouth et al., 2023; Wohltmann et al., 2024; Fleming et al., 2024). In January 2022 this submarine volcano injected huge amounts of water vapour directly into the middle stratosphere and even the lower mesosphere. Wang et al. (2023); Zhang et al. (2024a, b) focus on enhanced ozone depletion due to enhanced aerosol surface area and water vapour using satellite observations and the WACCM-model. Santee et al. (2023, 2024) provide further references and observational data and some review of involved heterogeneous and gas phase chemistry. Zhu et al. (2023) analyse ozone depletion inside the plume, a process that we do not treat in our study.

The main focus of this paper is on aerosol and heterogeneous chemistry in the lower stratosphere which has the largest impact on radiative forcing and total ozone. For the analyzed period 2017 to 2023 the Southern Hemisphere is of particular interest. With the chemistry climate model EMAC (ECHAM5-MESSy Atmospheric Chemistry) with extended heterogeneous chemistry on particles, introduced in section 2, we are able to reproduce most of the features observed by OSIRIS (Optical Spectrograph and InfraRed Imaging System), MLS (Microwave Limb Sounder), OMPS-LP (Ozone Mapping and Profiler Suite) and other satellite and balloon instruments and Lidar as shown in section 3. In this section we also discuss the radiative effects and stratospheric climate implications.

## 2 The chemistry climate model EMAC and used satellite data

### 2.1 EMAC and setup of the simulations

EMAC consists of the general circulation model ECHAM5 (Giorgetta et al., 2006) and the Modular Earth Submodel System (MESSy) V2.52 (Jöckel et al., 2010) including atmospheric chemistry. We use a setup in the spectral resolution T63 (1.9°) with 90 layers up to 1 Pa (80 km) where the meteorology of the troposphere (up to 100 hPa) is nudged to the reanalysis ERA5 (Hersbach et al., 2020) as recommended by Jöckel et al. (2006). Sea surface temperatures and sea ice cover are prescribed by ERA5. The Quasi Biennial Oscillation (QBO) is internally generated but slightly nudged to observations compiled by the Free University of Berlin and Karlsruhe Institute of Technology (see https://www.atmohub.kit.edu/english/807.php, last accessed June 2025) using a relaxation time of 2 months to allow for a comparison with satellite data (Giorgetta and Bengsston, 1999).

The model contains interactive gas phase and heterogeneous chemistry and the aerosol microphysics module GMXE (Pringle et al., 2010) with 4 soluble and 3 insoluble modes with parameters as in Brühl et al. (2018) and EQSAM aerosol chemistry

(Metzger and Lelieveld, 2007). Our module for heterogeneous chemistry on liquid stratospheric aerosol or polar stratospheric cloud (PSC) particles, MSBM, was modified as in Solomon et al. (2023), assuming that organic acids (e.g. hexanoic and acetic acids) in a particle strongly enhance the solubility of HCl in the particle compared to sulfuric acid or a mixture of sulfuric and nitric acid (Carslaw et al., 1995; Hanson et al., 1994; Huthwelker et al., 1995; Luo et al., 1995), accelerating the classical PSC reactions (Table 1) for chlorine activation also in midlatitudes. As in Solomon et al. (2023) the high solubility in hexanoic acid is applied if the mass ratio of organic carbon to sulfate exceeds 1. This leads to an HCl uptake and reactive chlorine release at much higher temperatures than for classical PSC or sulfate particles. The enhancement of surface area density (SAD) by organic aerosol leads also to faster conversion of NOx to $HNO_3$ via $N_2O_5 + H_2O \rightarrow 2\ HNO_3$. The simulation is a continuation

| Reaction | Solid PSC | Liquid PSC | Strat. aerosol |
|---|---|---|---|
| $N_2O_5 + H_2O \rightarrow 2\ HNO_3$ | yes | yes | yes |
| $ClONO_2 + H_2O \rightarrow HOCl + HNO_3$ | yes | yes | yes |
| $ClONO_2 + HCl \rightarrow Cl_2 + HNO_3$ | yes | yes | yes |
| $HOCl + HCl \rightarrow Cl_2 + H_2O$ | yes | yes | yes |
| $N_2O_5 + HCl \rightarrow ClONO_2 + HNO_3$ | yes | no | no |
| $BrONO_2 + H_2O \rightarrow HOBr + HNO_3$ | yes | yes | yes |
| $HOBr + HBr \rightarrow Br_2 + H_2O$ | yes | yes | no* |
| $ClONO_2 + HBr \rightarrow BrCl + HNO_3$ | yes | no | no |
| $BrONO_2 + HCl \rightarrow BrCl + HNO_3$ | yes | no | no |
| $HOCl + HBr \rightarrow BrCl + H_2O$ | yes | yes | no* |
| $HOBr + HCl \rightarrow BrCl + H_2O$ | yes | yes | no* |

**Table 1.** Heterogeneous reaction on PSC particles, sulfate and organic aerosol (* yes in a sensitivity study based on scenario 6 of Table 2).

of the multidecadal transient simulation of Schallock et al. (2023) including several hundred explosive volcanic eruptions derived after March 2012 from OSIRIS extinction data. Kohl et al. (2025) contains a continuation of the volcanic emission inventory from September 2019 to December 2023. For the powerful eruption of Hunga in January 2022 we assume that 400 kt $SO_2$ (Zhu et al., 2022) was injected as a mixing ratio perturbation near 25 km altitude based on the 3D aerosol plume seen by OSIRIS. A sensitivity study with 500 kt $SO_2$ and a slightly higher plume top height was also performed. The water vapour injection of 136 Mt by Hunga at 13-36 hPa is considered with the module TREXP (Jöckel et al., 2010; Kohl et al., 2025) assuming a Gaussian vertical distribution centered at 21.5 hPa and truncated at 2.4 $\sigma$ with $\sigma$=1.25 km, and a slab covering 4 neighboring horizontal grid boxes, i.e. in total 48 boxes. Note that our estimate based on preliminary data is slightly less than the 146 Mt proposed by Millán et al. (2022) but the problem was that only 64 Mt were retained in the stratosphere because of removal by ice formation. To improve this we performed a sensitivity study with 150 Mt $H_2O$ and 500 kt $SO_2$ injected in 9 neighboring grid boxes and the peak for the Gaussian distribution for the $H_2O$ injection at 17hPa. Here 122 Mt of $H_2O$ were retained. To study the microphysical, chemical and radiative interactions also a simulation with the same $SO_2$ injection but no water vapour injection is included. The performed simulations are summarized in Table 2.

| Scenario | Fire aerosol | Het. chem. on organics | Hunga $SO_2$ method or box and peak, mass | Hunga $H_2O$ box, peak of Gaussian, mass | color |
|---|---|---|---|---|---|
| 1 | no | no | OSIRIS, 400kt | $3.8 \times 3.8^o$, 21hPa, 136Mt | green |
| 2 | yes | no | OSIRIS, 400kt | $3.8 \times 3.8^o$, 21hPa, 136Mt | blue |
| 3 | yes | no | no | no | purple |
| 4 | yes | yes | OSIRIS, 400kt | $3.8 \times 3.8^o$, 21hPa, 136Mt | red |
| 5 | yes | yes | OSIRIS, 500kt | $3.8 \times 3.8^o$, 21hPa, 136Mt | red dashed |
| 6 | yes | yes | $5.7 \times 5.7^o$, 21hpa, 500kt | $5.7 \times 5.7^o$, 17hPa, 150Mt | lightblue |
| 7 | yes | yes | $5.7 \times 5.7^o$, 21hpa, 500kt | no | lightblue dashed |

**Table 2.** The 7 transient simulations with the color of the lines in the figures. Scenarios 5 to 7 differ from scenario 4 only after 15 January 2022.

Organic and black carbon (OC and BC) from the 3 major forest fires in 2017 to 2020 are injected in the Aitken mode at the top of the pyro-cumulonimbi into the lower stratosphere (1.5 km thick boxes) using TREXP with data from Peterson et al. (2018) for the British Columbia fire in August 2017, Osborne et al. (2022) for the Alberta fire in June 2019 and Peterson et al. (2021) for the Australian fire in December 2019 / January 2020. For the British Columbia fire we considered 4 3h events on August 12/13 emitting in total 300 kt OC and 19.5 kt BC in 4 slabs into the 132-175 hPa layer at slightly different locations

and times near $51^oN/122^oW$. The smoke of the Alberta fire is injected on June 17 in the same altitude range into 4 neighboring slabs near $60.5^oN/116^oW$ with in total 108 kt OC and 7 kt BC. The Australian fire was modeled as a 63 h lasting event starting on December 29 injecting 600 kt OC and 39 kt BC into the layer between 92 and 121 hPa covering 4 horizontal gridboxes around $37^oS/149^oE$, followed by a 6 h event on January 4 injecting 216 kt OC and 13.8 kt BC into the same slabs. Organic carbon is assumed to be in the soluble and black carbon in the insoluble mode. The absorbing smoke particles experience

lofting by radiative heating reaching about 28 km in case of the Australian fire.

## 2.2 OSIRIS

OSIRIS is a limb scatter observing instrument, which was launched on board the Odin satellite on 20 February 2001 and is still operating today. OSIRIS provides coverage from $82^oS$ to $82^oN$ over the course of the year. To obtain the vertical profiles of aerosol extinction at altitudes from 10 to 35 km, the aerosol scattering properties are calculated with a refractive index of

$1.427+i7.167\times10^{-8}$ using Mie theory at 750 nm wavelength and a sulfate concentration of 75% $H_2SO_4$ and 25% $H_2O$ (Rieger et al., 2019). Extinction is retrieved where the tangent point is illuminated, which is primarily in the summer hemisphere, using version 7.3. The grid resolution is 1 km altitude, $5^o$ latitude, and $30^o$ longitude with 5 day averaged time intervals. Observed zonal average stratospheric aerosol optical depth (SAOD) for 3 latitude ranges and 5 years is shown by the black dashed curves in Fig. 1.

## 2.3 OMPS-LP

The OMPS-LP instrument on the Suomi National Polar Orbiting satellite, which was launched in 2011, measures vertical images of the spectral radiance of the atmospheric limb. These scattered-sunlight spectra, in the range 290–1,000 nm, are used in combination with a radiative transfer forward model to retrieve vertical profiles of ozone number density and aerosol extinction coefficient. Measurements are made along the daylight portion of the satellite orbital track. Aerosol extinction is retrieved at a single wavelength channel of 746 nm, derived using a two-dimensional, or tomographic, inversion (Rieger et al., 2021; Bourassa et al., 2023). The vertical and horizontal resolutions of the retrieval are approximately 1.5 and 250 km, respectively, and profiles generally extend from the tropopause through the upper stratosphere. Here we use L2-version 1.3, the grid resolution is like for the OSIRIS data set. Zonal average observations of SAOD are shown by the black dotted curves in Fig. 1.

## 2.4 GloSSAC satellite climatology

The Global Space-based Stratospheric Aerosol Climatology (GloSSAC, V2.22, Kovilakam et al. (2023), Knepp et al. (2024)) contains observed multi-satellite monthly aerosol extinctions from 1979 to 2023 and is based for the considered period 2017 to 2023 mostly on SAGE III/ISS (Stratospheric Aerosol and Gas Experiment on International Space Station) data in tropics and midlatitudes ($< 60^o$). We used the cloud cleared and the full SAGE III/ISS datasets (Black thick and black dash-dot lines in Fig. 1).

## 2.5 AURA-MLS

MLS onboard NASA's Aura satellite provides measurements of 15 trace gases, among them $H_2O$, HCl, ClO, $HNO_3$ and $O_3$ we use for evaluation. Here we use version 5 following Santee et al. (2023). MLS measures thermal emission from the Earth's limb, covering spectral regions near 118, 190, 240, and 640 GHz. We use zonal average daily gridded L3 data on pressure levels with a resolution of $4^o$ in latitude and about 2.4 km in altitude. Compared to other observations, MLS HCl can be up to 10% higher (Livesey et al., 2022).

## 3 Results for 2017-2023

We performed a transient simulation without forest fires, simulations with the radiative and dynamical effects of the forest fires mentioned in section 2, with and without the eruption of Hunga, and simulations with heterogeneous chemistry on organic aerosol for the standard Hunga case and 3 sensitivity studies with more $SO_2$ including one with a more realistic injection of Hunga water vapor addressed above and summarized in Table 2.

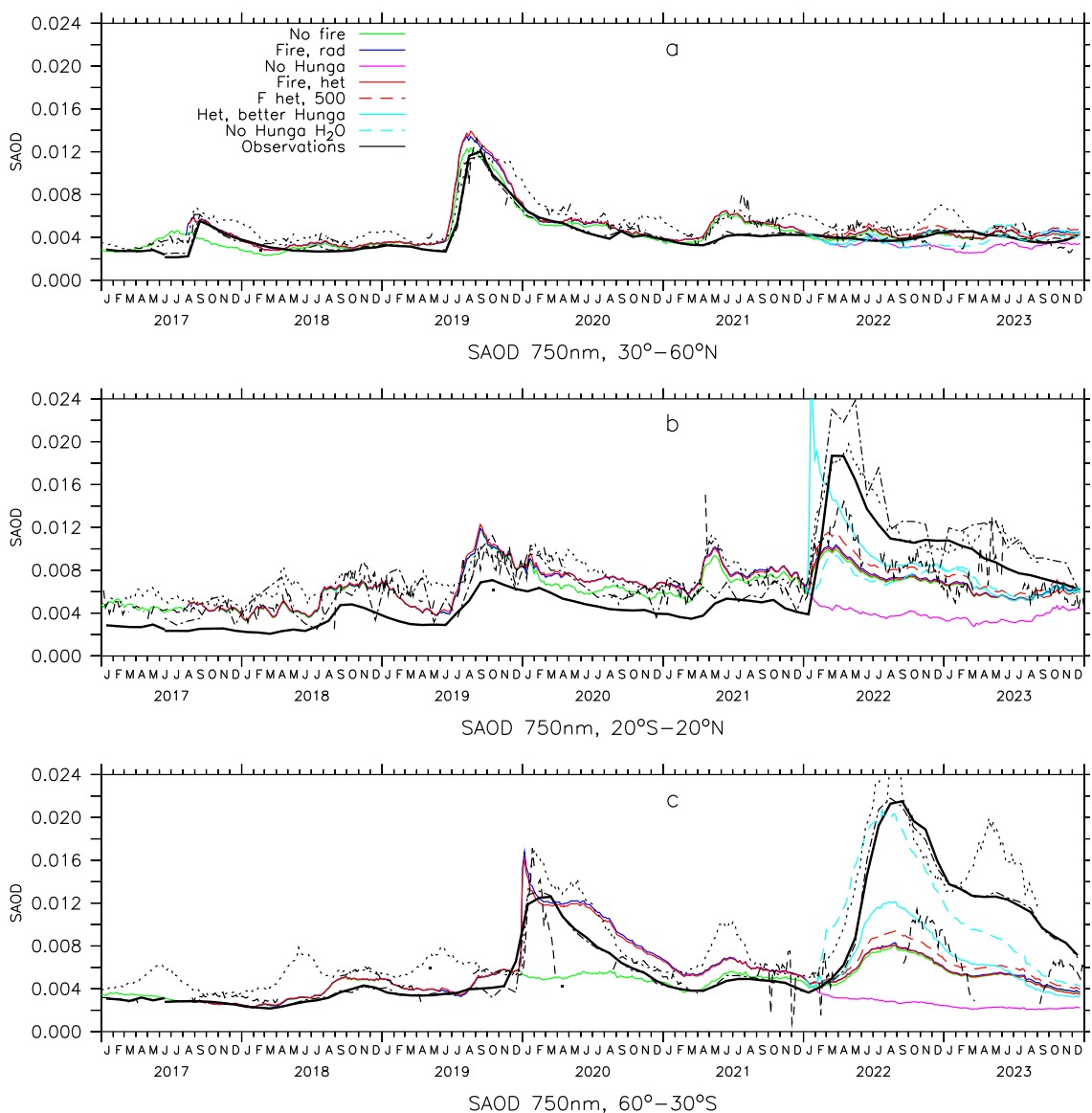

**Figure 1.** Stratospheric aerosol optical depth in tropical latitudes (b), northern (a) and southern midlatitudes (c), zonal averages. Colors: EMAC simulations of legend and Table 2. Black thick: GloSSAC V2.22, after June 2017 observations by SAGE III/ISS, cloud cleared, black dash-dotted: GloSSAC V2.22, SAGE III/ISS, all, black dotted: observed by OMPS-LP, black dashed: observed by OSIRIS. GloSSAC monthly averages, all other curves 5-day averages.

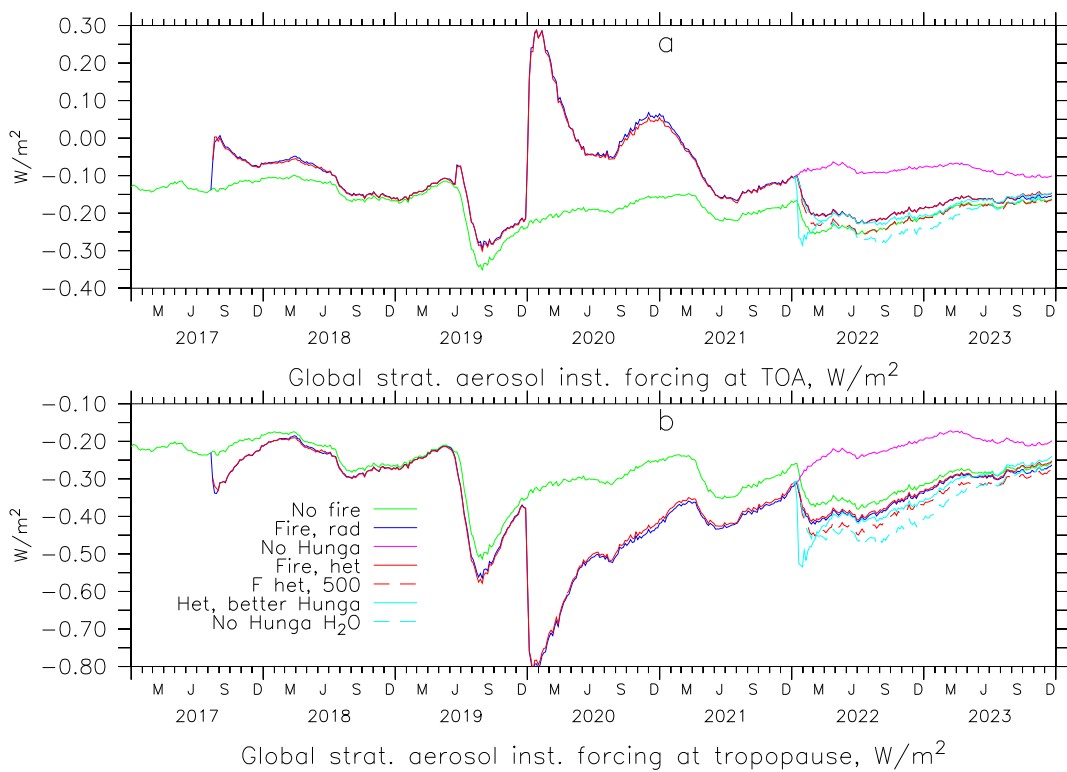

**Figure 2.** Calculated instantaneous radiative forcing by aerosol at the Top of the Atmosphere (TOA, a) and tropopause altitude (b). Colors of EMAC simulations see legend and Table 2.

## 3.1 Optical depth, radiative forcing and heating

Simulated and observed stratospheric aerosol optical depth (SAOD) for the tropics, the southern and the northern midlatitudes is presented in Fig. 1. Including the organic aerosol from major forest fires is essential for agreement with the observations by SAGE III/ISS, OSIRIS and OMPS-LP. In southern midlatitudes the Australian forest fire enhances SAOD in January 2020 in observations and simulations by about 0.011 in zonal mean (black, red and green curves). Two years later the perturbation is still 0.0005 as can be seen by the difference between the red and green curves, consistent also with lidar observations by Ohneiser et al. (2022). In northern midlatitudes, SAOD is dominated by the eruption of Raikoke in 2019 (0.009) while Canadian fires cause a slight enhancement (0.001 in 2019 and 0.002 in 2017). The eruption of Hunga in January 2022 dominates SAOD in the next years with a contribution of up to 0.018 in tropics and southern midlatitudes. In the tropics, the eruptions of Ulawun in June and August 2019 and of St. Vincent in April 2021 have also a large impact on SAOD (0.008 and 0.004, respectively), while the effect of Australian fires is relatively small, up to 0.0013. Note that the very low OSIRIS values near the data gaps in midlatitudes are artifacts due to sparse coverage, values at very large solar zenith angles and often missing observations of the lowermost stratosphere and should be ignored. Also, this feature does not happen in OMPS-LP observations (Rieger et al.,

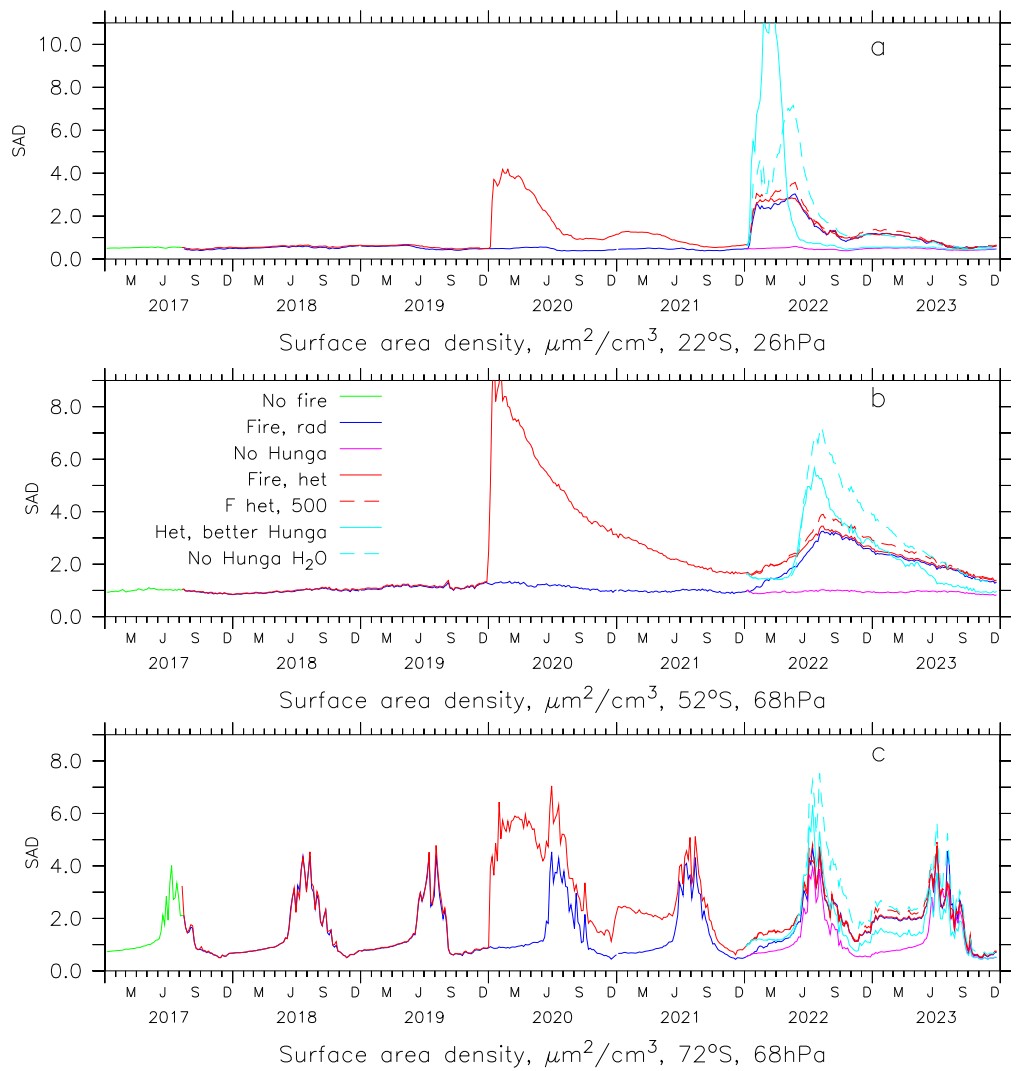

**Figure 3.** Calculated surface area density [$\mu$m$^2$/cm$^3$] for heterogeneous chemistry in the middle stratosphere in southern subtropics (a) and the lower stratosphere in southern middle (b) and high (c) latitudes. Colors see legend and Table 2. The green curve is shown only for 2017 since very close to the blue one later.

2021; Bourassa et al., 2023) and GloSSAC (Kovilakam et al., 2023). However, OMPS-LP observations (dotted black lines) sometimes appear to be high-biased compared to other instruments. During northern summer 2020 OSIRIS data were so sparse that OMPS-LP had to be used to estimate SO$_2$ injections of some volcanic events and it looks like that the used correction factors "f" by Schallock et al. (2023) for these were too large, leading to an overestimate of the sulfate contribution to extinction

still visible to the end of 2020 in the southern hemisphere. In April 2023 there appears to be a southern midlatitude volcanic event missing in our inventory but it is more likely that the difference to observations is due to the not well captured impacts of Hunga emission patterns on model dynamics. One has also to keep in mind that OMPS-LP appears to be high-biased relative to other instruments in local fall in both hemispheres. Also in April 2023 some source of aerosol appears to be missing in the tropics but this is most likely an artifact in the satellite data due to high clouds since this does not appear in the cloud cleared SAGE III/ISS data. Sensitivity study 5 with enhanced $SO_2$ injection by Hunga improves the agreement with OSIRIS and SAGE III/ISS observations in late 2022, but still underestimates SH midlatitude SAOD in 2023, especially when comparing to GloSSAC (thick black lines). Ignoring Hunga reduces SAOD by up to 0.0060 to 0.0075. In scenario 6 where $SO_2$ is almost co-injected with $H_2O$ in a $5.7 \times 5.7^o$ region (see Table 2) the strongly enhanced OH production leads to a very quick sulfur aerosol formation causing the too early peak of SAOD in the tropics. Radiative heating of this aerosol causes a lofting of the plume and a faster spread to midlatitudes leading to a better agreement of SAOD with GloSSAC there. A problem with this scenario is that the fast growth of aerosol particle due to water uptake cause a too fast sedimentation already near the source region so that scenario 7 (lightblue dashed curves) without $H_2O$ injection reproduces observed aerosol optical depth much better, including the slow decrease during Austral fall of 2023.

The global instantaneous stratospheric aerosol forcing calculated online by multiple radiation calls at the top of the atmosphere (TOA) and in the altitude of the midlatitude tropopause is shown in Fig. 2. The Australian fire emissions change the TOA forcing by up to $0.5 \, \mathrm{W\,m^{-2}}$ in January 2020 with the perturbation modulated by insolation lasting more than 3 years with values of $0.07 \, \mathrm{W\,m^{-2}}$ in January 2022 and $0.02 \, \mathrm{W\,m^{-2}}$ in January 2023. The signals of $0.11 \, \mathrm{W\,m^{-2}}$ in August 2017 and $0.05 \, \mathrm{W\,m^{-2}}$ after June 2019 are related to Canadian fires. The sign change of the forcing perturbation by smoke with altitude is due to its light absorbing properties. The computed global aerosol radiative forcing at TOA caused by the Hunga eruption in 2022 was about $-0.12 \, \mathrm{W\,m^{-2}}$, decreasing to $-0.06 \, \mathrm{W\,m^{-2}}$ by December 2023, dominated by the change in stratospheric sulfate aerosols. In the sensitivity simulation 5 with more Hunga $SO_2$ these numbers change to $-0.15 \, \mathrm{W\,m^{-2}}$ and $-0.07 \, \mathrm{W\,m^{-2}}$. In the experiment 6 with almost co-injection of $SO_2$ and $H_2O$ the effect of Hunga is larger in the first 2 months and smaller later, approaching scenario 5. Surprisingly, the largest forcing by Hunga occurs in no water scenario 7 with $-0.17 \, \mathrm{W\,m^{-2}}$ in August 2022.

Solar and infrared radiative heating rates by smoke and volcanic aerosol in tropics and midlatitudes in Figs. A6 and A7 are largest in the middle stratosphere with up to 0.16 K/d at 28 km in the tropics and 0.36 K/d at 20 km in midlatitudes in zonal average. The heating by smoke from the Australian fires leads to a zonal average temperature increase of up to about 3 K over several months in large parts of the SH lower stratosphere. This feature is similar to that reported by Yu et al. (2021). The warmer tropopause region also causes an increase in stratospheric water vapour (Fig. A1). Hunga aerosol causes a heating rate by up to 0.16 K/d at 25 km in the tropics and 0.05 K/d in southern midlatitudes (panels a and b of Figs. A6 and A7). The cooling rate by the injected water vapour cannot be directly derived using our online method. The heating by up to 0.032 K/d in the tropics at 18 km is related to the eruptions of Ulawun and St. Vincent (panel c).

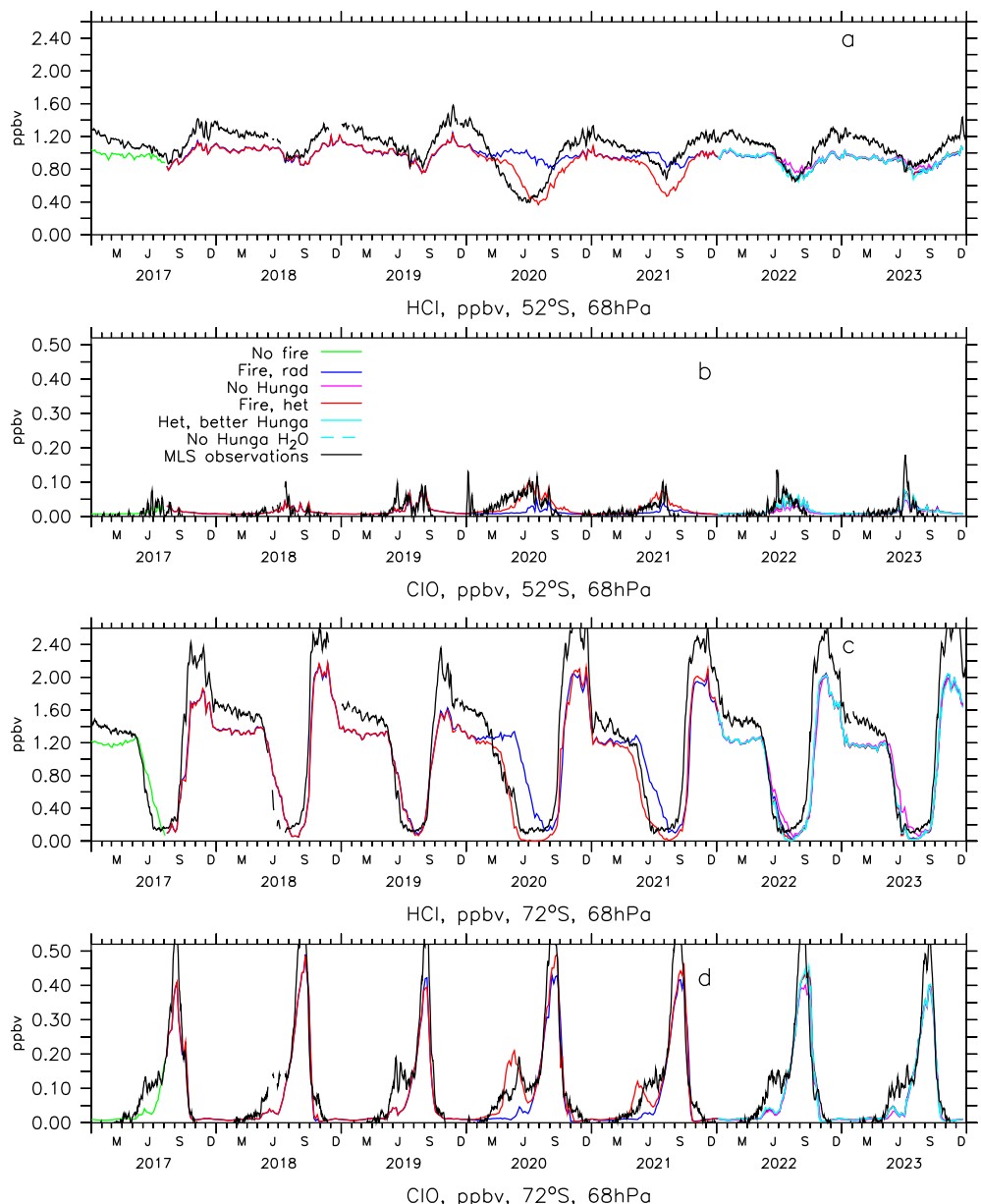

**Figure 4.** Simulated and observed HCl at 68 hPa at 52°S (a) and 72°S (c). Calculated and observed ClO in panels (b) and (d). Colors in the legend indicate the EMAC scenarios of Table 2. 5-day averages for EMAC, daily averages with 5-day boxcar smoothing for MLS.

### 3.2 Perturbation of dynamics by smoke

In case of the Australian fires the strong radiative heating in the absorbing aerosol containing plumes causes their lofting and the formation of anticyclonic vortices. The vortices generated by EMAC are similar in size as the observed ones discussed in Kablick et al. (2020) but the location differs and the lofting is somewhat faster. EMAC reproduces the reduced ozone and enhanced $N_2O$ in the smoke filled vortices due to vertical displacement. An example for January 2020 and southern midlatitudes is shown in Figs. A8 and A9. These perturbations are visible for about 2 months. In the used setup of the model the vortices rise only to about 28 km, the maximum altitude where heating by aerosol is considered in the radiation scheme. Because of this limitation the perturbations found by Ma et al. (2024) above this altitude could not be studied.

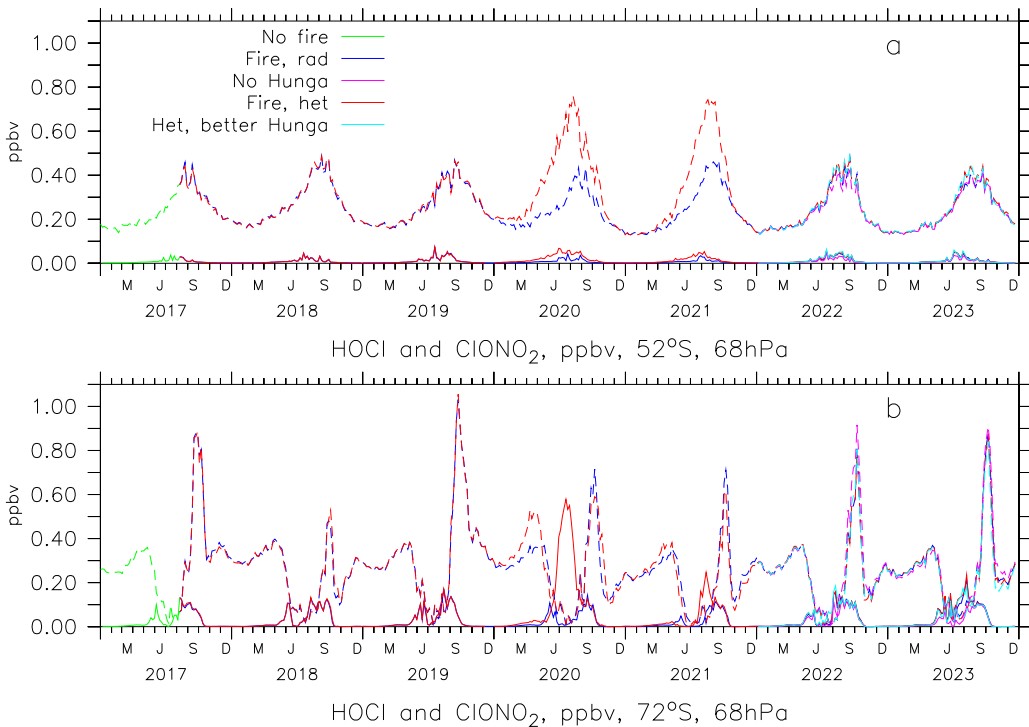

**Figure 5.** Simulated HOCl at 68 hPa at 52$^o$S (a) and 72$^o$S (b). Full curves, shown EMAC scenarios of Table 2 in legend. Dashed curves show calculated ClONO$_2$. Scenario 5 and 7 not shown.

### 3.3 Aerosol properties and ozone chemistry

Calculated number densities N and median wet radii $r_{median}$ in the accumulation mode in tropics and SH midlatitudes are presented in Figs. A4 and A5. N increases by up to about 50% for the Australian fire and by up to 100% for Hunga at 52$^o$S and 68 hPa, for $r_{median}$ the increase there is about 0.03 $\mu$m for both perturbations.

The simulated particle surface area density (SAD) for use with heterogeneous chemistry is shown in Fig. 3. The organic particles injected into the stratosphere by the Australian fires cause a large enhancement of SAD by up to about a factor of 10 in the whole Southern hemisphere. In the calculation we assume here the molecular mass of acetic acid. Hunga is responsible for about a doubling to tripling of SAD outside the regions with polar stratospheric clouds (PSC, causing the winter peaks) at 68 hPa.

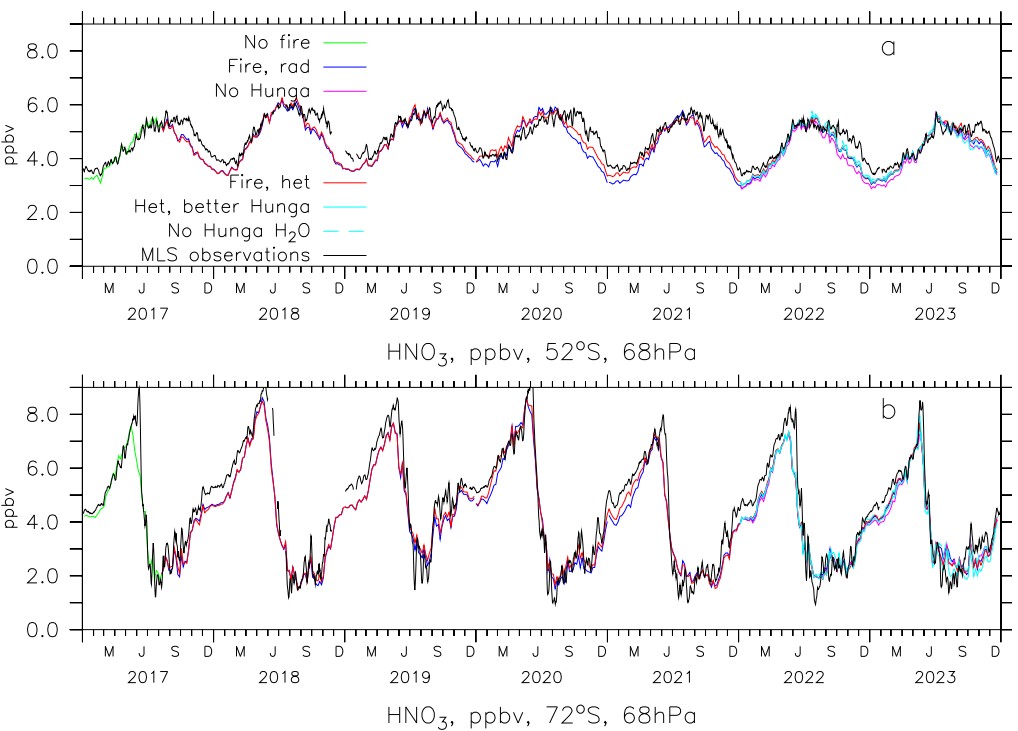

**Figure 6.** Simulated and observed HNO$_3$ at 68 hPa at 52$^o$S (a) and 72$^o$S (b). EMAC scenarios of Table 2 see legend.

Simulated and observed HCl and ClO for mid and high southern latitudes are depicted in Fig. 4, simulated HOCl and ClONO$_2$ in Fig. 5. It is clearly seen that heterogeneous chlorine activation on organic particles has to be included to reproduce the HCl observations by MLS with a fast decrease in Austral fall and early winter of 2020 and 2021 (red and black curves) accompanied by an increase in ClO as also found by Solomon et al. (2023) and Stone et al. (2025) for both latitude regions in 2020. In 2021 this perturbation is larger than in Stone et al. (2025). The time of the HCl recovery depends strongly on how much local ozone is left and near the vortex edge model uncertainties on this are largest (Grooß et al., 2011). In the presence of organic aerosol in these winters ClONO$_2$ is strongly enhanced in midlatitudes by up to 0.4 ppbv in agreement with Solomon et al. (2023) and in April and September in high latitudes. The latter causes a delay in HCl recovery compared to MLS. HOCl is strongly enhanced in high latitude winter of 2020. In Northern high latitudes in early spring of 2020 the model simulates a

slight chlorine activation due to the 2019 Canadian fires not shown here. The 2017 Canadian fire causes a ClO increase in early Arctic winter but no significant ozone change.

Heterogeneous chlorine activation on Hunga aerosol is important in Austral winters of 2022 and 2023 (blue/red curves against purple ones). Here we obtain similar results as Zhang et al. (2024b), we found however from a sensitivity study based on scenario 6 of Table 2 with the bromine reactions in Table 1 marked with * active in midlatitudes, that the reaction HOBr+HCl contributes only a small fraction to chlorine activation (0.02 ppbv HCl decrease).

   For the nitrogen species mostly the effect of the SAD increase due to smoke and volcano matters, causing an increase in
$HNO_3$ (Fig. 6) and a decrease in $NO_2$ (Fig. A2). For $HNO_3$ the EMAC simulations match almost perfectly with the MLS observations if the heterogeneous chemistry on smoke aerosol is included (Fig. 6). The reduction of NOx by heterogeneous reactions on organic and sulfate aerosol leads to an increase in the hydroxy radical due to $NO + HO_2 \rightarrow NO_2 + OH$ (Fig. A3, red against blue or purple curve). This effect and the Hunga water vapour injections and the subsequent increase in stratospheric water vapour (Fig. A1) cause an enhancement in $HO_2$ (Fig. A3) by almost 20% in Austral summers of 2022/2023 and 2023/2024.

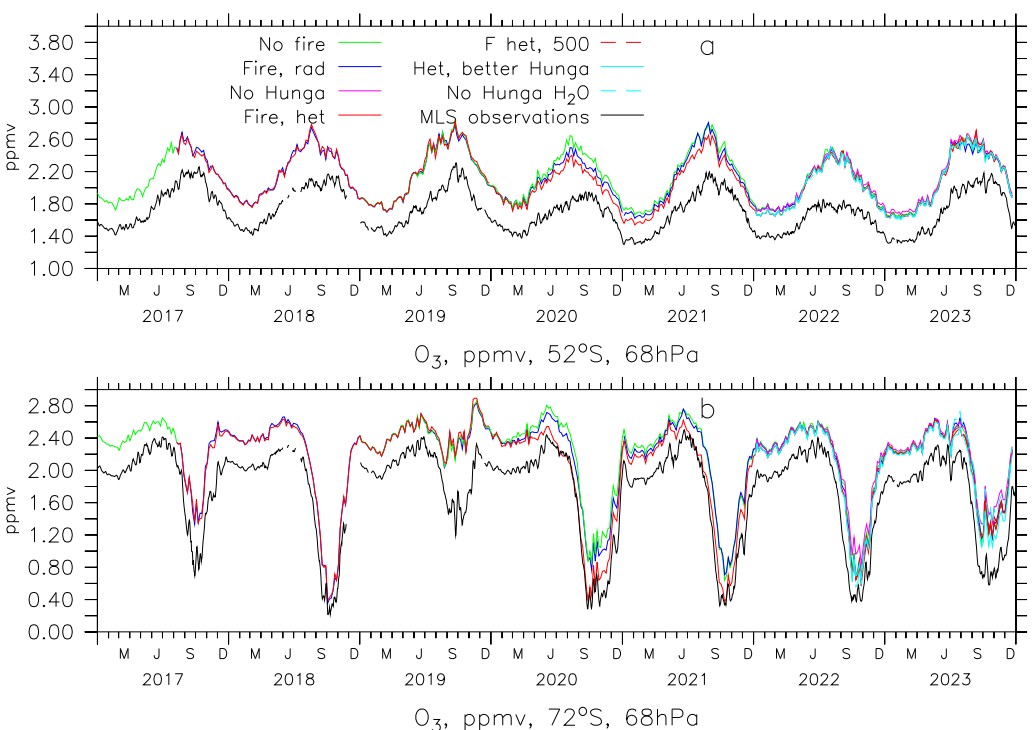

**Figure 7.** Simulated and observed $O_3$ at 68 hPa at 52$^o$S (a) and 72$^o$S (b). EMAC scenarios of Table 2 see legend.

   Simulated and observed ozone is shown in Fig. 7. The high bias in EMAC lower stratospheric ozone is most likely caused by numerical diffusion and to a small fraction by the underestimated water vapour (Fig. A1) related to a cold bias at the tropical

tropopause. In years with weak vortices, numerical diffusion can cause an underestimate of ozone depletion, which is not the case in years with strong vortices (e. g. 2018). Nevertheless, relative effects from fire and volcano emissions can be studied. Here the combined effect of perturbation of dynamics and chemistry by smoke causes a zonal average decrease of almost 10% in the midlatitude lower stratosphere, almost complete loss at the edge of the ozone hole and complete loss south of $80^o$S (not shown) in winter 2020 and 2021 in agreement with MLS (black and red curves). In contrast, the remaining ozone in the control run (green) and the run without heterogeneous chemistry on organics (blue) is 0.2 ppmv ($80^o$S) to 0.5 ppmv ($72^o$S) higher. In 2020 the calculated ClO increase near the edge of the Antarctic vortex occurs earlier than in the observations (Fig. 4d) leading to the earlier ozone decrease in Austral fall which reduces the bias between 60 and $80^o$ S. The effect of Hunga is mostly visible at the vortex edge in October 2022 (b, purple and blue or lightblue curves). Simulated ozone changes are consistent with Solomon et al. (2023), Stone et al. (2025) for the wildfires and Zhang et al. (2024a) for Hunga.

Heterogeneous chlorine activation causes decreases in zonally averaged total ozone of more than 30 DU in the first 2 years after the Australian fires (Fig. 8a), Changes in dynamics due to heating by smoke cause a total ozone decrease by up to about 24 DU (Fig. 8d), combined effects add up to a loss of up to 45 DU (scenario 4 against scenario 1). At $50^o$S the typical decrease in Austral winter still exceeds 10 DU. The loss due to the eruption of Hunga is up to about 15 DU near the edge of the Antarctic vortex in 2022 and 2023 (Fig. 8b) and some DU more in the first 6 months of 2022 in case of scenario 6 (Fig. 8c).

Spatial co-injection of $SO_2$ and $H_2O$ alters the results since it can prevent the descent of the $H_2O$ plume by radiative cooling in the early phase due to heating of the sulfur aerosol which is formed too rapidly, related to strongly enhanced OH. This leads to a too late arrival of the additional water vapour in the lower stratosphere at mid and high southern latitudes. Additionally, too big particles are formed via coagulation, causing an overestimation of removal by sedimentation near the source region. However, some overlap of the $H_2O$ and $SO_2$ plumes needs to be assumed to explain the observed fast $SO_2$ conversion (Santee et al., 2023). Surprisingly, in our model observed SAOD is best reproduced in scenario 7 where Hunga water vapor is ignored. The simulated impact of the water vapor injection on the aerosol size distribution near the source region is consistent with Li et al. (2024) (Fig. A10 a,b). Fig. A10 shows also that Hunga water causes an ozone increase at the latitude and altitude of the injection, despite enhanced chlorine activation, because of the strongly altitude dependent interaction between the different ozone depleting catalytic cycles.

## 4   Conclusions

The Australian forest fires in 2019/2020 caused the largest perturbation in stratospheric optical depth (for example, in OSIRIS and OMPS-LP data) and instantaneous radiative forcing since the eruption of Pinatubo by up to 0.5 $W\,m^{-2}$ at the top of the atmosphere and -0.4 $W\,m^{-2}$ at the tropopause. In January 2022, the remaining effect at TOA was about 0.07 $W\,m^{-2}$, still counteracting the negative forcing by volcanoes, which was in case of Hunga up to -0.16 $W\,m^{-2}$. These fires caused a zonal average heating of the lower stratosphere of the Southern Hemisphere by up to about 3 K for several months. The absorbing aerosol from the Boreal fires in 2017 and 2019 reduced the volcanic forcing at TOA but enhanced it at the tropopause by 0.11 $W\,m^{-2}$ and 0.05 $W\,m^{-2}$, respectively. It is needed to explain the observed SAOD.

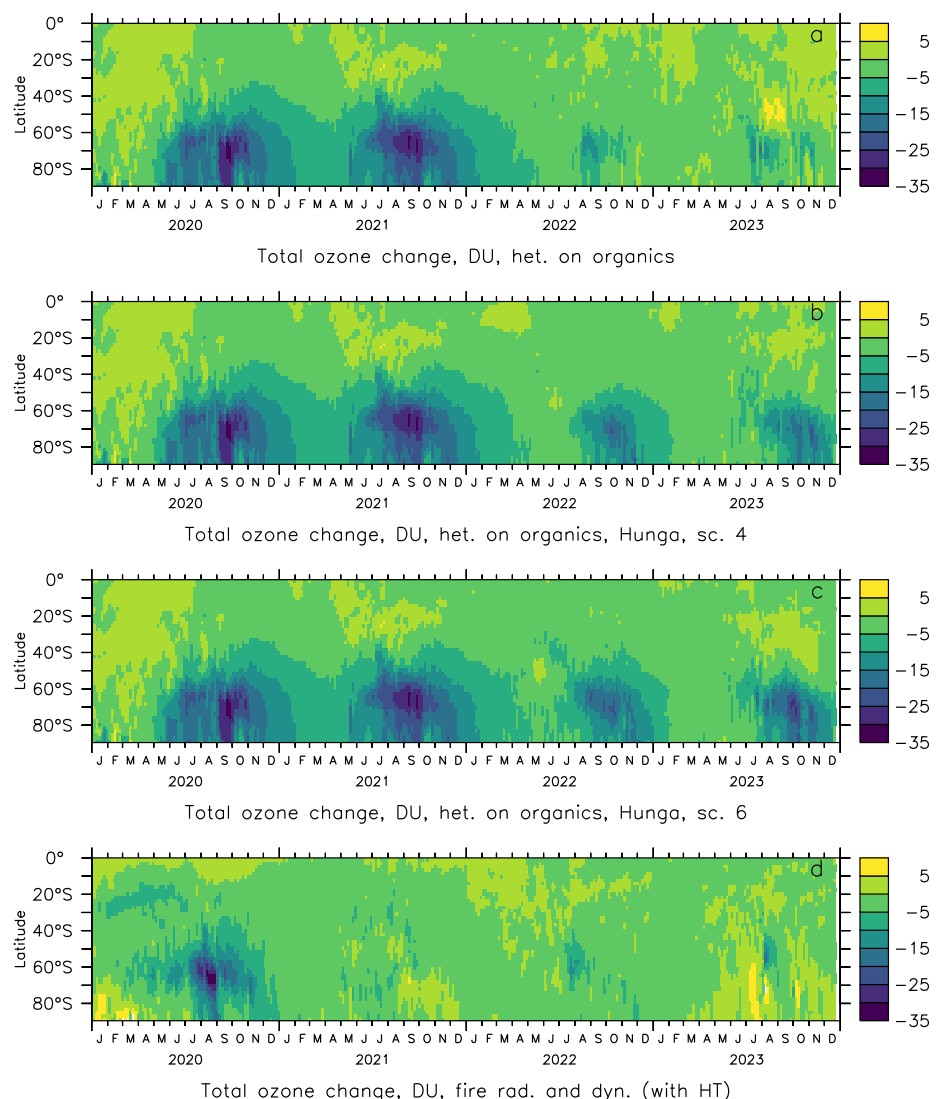

**Figure 8.** Calculated total ozone change from heterogeneous chlorine activation on organics from fires (a, scenario 4 against scenario 2) and additionally the eruption of Hunga (scenario 4 (b) and scenario 6 (c) against scenario 3). The lower panel (d) shows the radiative/dynamical effects of fires (scenario 2 against scenario 1, in 2023 mostly noise).

To obtain agreement with the chlorine activation in polar regions and midlatitudes observed by MLS it is essential to include the enhanced solubility of HCl in particles containing organic acids from major forest fires, in agreement with Solomon et al. (2023); Stone et al. (2025). The Australian fire emissions led to enhanced ozone depletion in fall, winter and spring in high and

mid latitudes reducing minimum total ozone in 2020 and 2021 by up to 30 DU around $70^{o}$S, accompanied by a reduction in August 2020 from the lofting smoke filled vortices and other perturbations in dynamics by up to 24 DU (total up to about 45 DU).

The total ozone loss due to the eruption of Hunga was up to about 15 DU near $70^{o}$S in 2022 and 2023. Hunga water vapour had only a small effect on ozone and radiative forcing. EMAC reproduces the water vapour increase seen by MLS if injected in a large enough volume and a vertical distribution shifted upward by about 1.5 km compared to $SO_2$ to avoid too much ice formation but has a low bias of almost 1 ppmv (outside of the Antarctic vortex, Fig. A1). For ozone the water vapour increase causes a change in the vertical distribution with almost no effect on total ozone, while the positive radiative forcing is not significant in the model. There is, however, an impact via the modified aerosol residence time. Our simulations show an extreme sensitivity of aerosol properties and radiative and chemical implications to the spatial distribution of the injections of Hunga $SO_2$ and water vapour.

We demonstrated that EMAC (in this version) is a useful tool for simulating the implications of major forest fires and volcanic eruptions on climate and stratospheric ozone, including the delay in the recovery of the Antarctic ozone hole.

*Code and data availability.* The Modular Earth Submodel System (MESSy) is continuously developed and used by a consortium of institutions. The use of MESSy and access to the source code is licensed to all affiliates of institutions which are members of the MESSy Consortium. Institutions can become a member of the MESSy Consortium by signing the MESSy Memorandum of Understanding. More information can be found on the MESSy Consortium website (http://www.messy-interface.org, last accessed June 2025). The namelists, input data files and model output of EMAC used here are stored at DKRZ, Hamburg, Germany. The frozen version of the used code in V2.52 is available from the authors on request.

The complete OSIRIS L2 data set can be downloaded from https://arg.usask.ca/docs/osiris_v7/, the OMPS-LP L2 aerosol extinction data from https://doi.org/10.5281/zenodo.7293121 (gridded data for both instruments can be requested from the authors). AURA-MLS L3 data (V5, gridded on pressure levels, daily) are available from Goddard Earth Sciences Data and Information Services Center (GES DISC, https://data.gesdisc.earthdata.nasa.gov/data/Aura_MLS_Level3). GloSSAC V2.22 is available from NASA at https://doi.org/10.5067/GLOSSAC-L3-V2.22.

*Author contributions.* CB modified the code, performed the simulations and wrote most of the paper, MK contributed to the setup of the volcano and wild fire emissions and the text, JL worked on the text.

*Competing interests.* The authors declare that they have no conflicts of interest.

*Acknowledgements.* We thank Landon Rieger, now at Canadian Centre for Climate Modelling and Analysis (CCCma), Environment and Climate Change Canada (ECCC), Victoria, BC, Canada, for providing OSIRIS and OMPS-LP data in the early phase of the paper, Adam Bourassa, University of Saskatchewan, Saskatoon, Canada, for advices to the OSIRIS and OMPS-LP data, and Michelle Santee, Jet Propulsion Laboratory, California Institute of Technology, Pasadena, CA, USA for help with MLS data and useful suggestions for the introduction.

AGAGE observations are used as lower boundary condition for long lived gases in EMAC. The computations were performed on the Levante supercomputer at DKRZ Hamburg, Germany, with support by Max Planck Society.

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

## Appendix A: Additional figures

This section provides the comparison of water vapour simulated by EMAC with MLS observations (Fig. A1) and simulated $NO_2$ (Fig. A2) and $HO_2$ (Fig. A3) as rate limiting radicals for catalytic ozone destruction. Further, number density (Fig. A4) and median radius (Fig. A5, for effective radius multiply by 1.49) for the accumulation mode of stratospheric aerosol are shown, as well as total radiative (solar+thermal) aerosol heating rates (Figs. A6 and A7) and examples for self lofting smoke filled vortices and the corresponding ozone reduction calculated by EMAC (Figs. A8 and A9). Fig. A10 shows microphysics

and chemistry for the latitude and altitude region of the Hunga injection (see also Figs. 3 and A1).

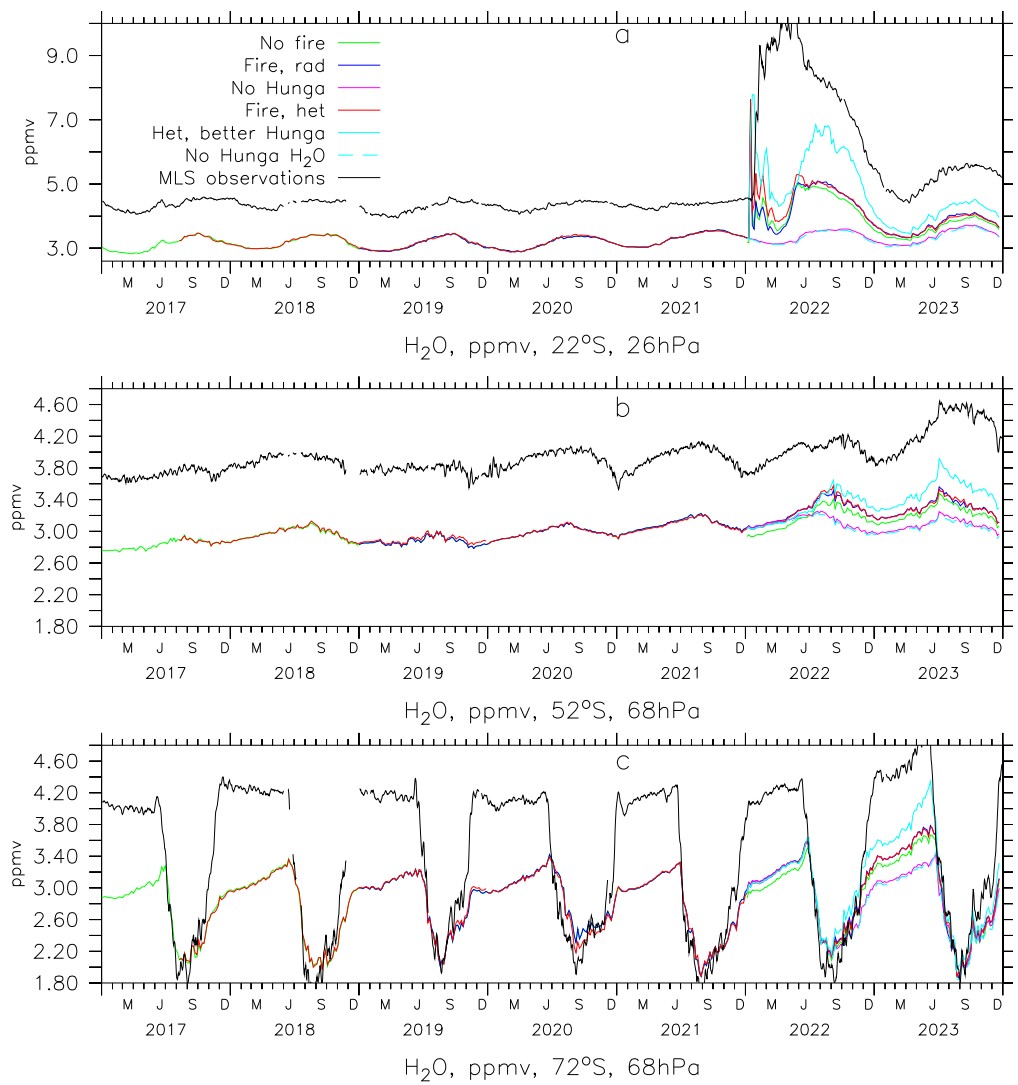

**Figure A1.** Calculated and observed water vapor, colors as in Fig. 7 (scenario 5 not shown). Panel (a) is for the subtropical middle stratosphere, (b) for the southern midlatitude lower stratosphere, (c) for southern high latitudes.

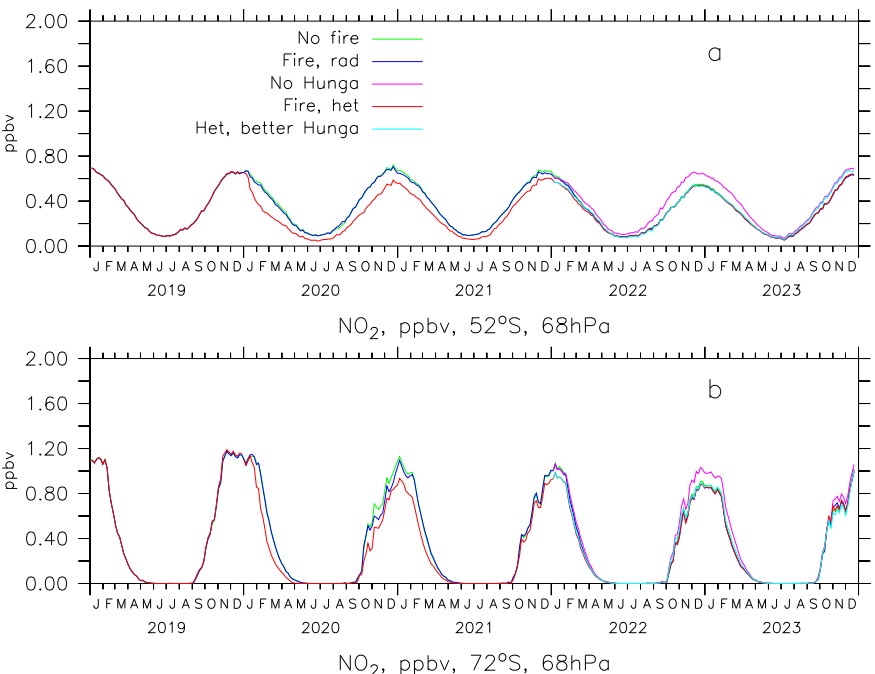

**Figure A2.** Calculated $NO_2$, colors as in Table 2.

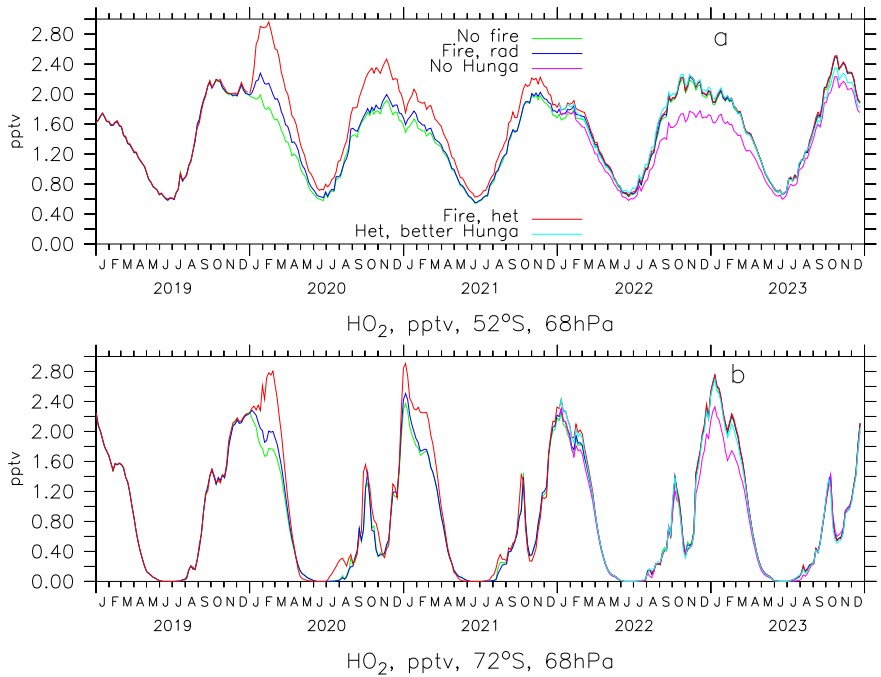

**Figure A3.** Calculated $HO_2$, colors as in Table 2.

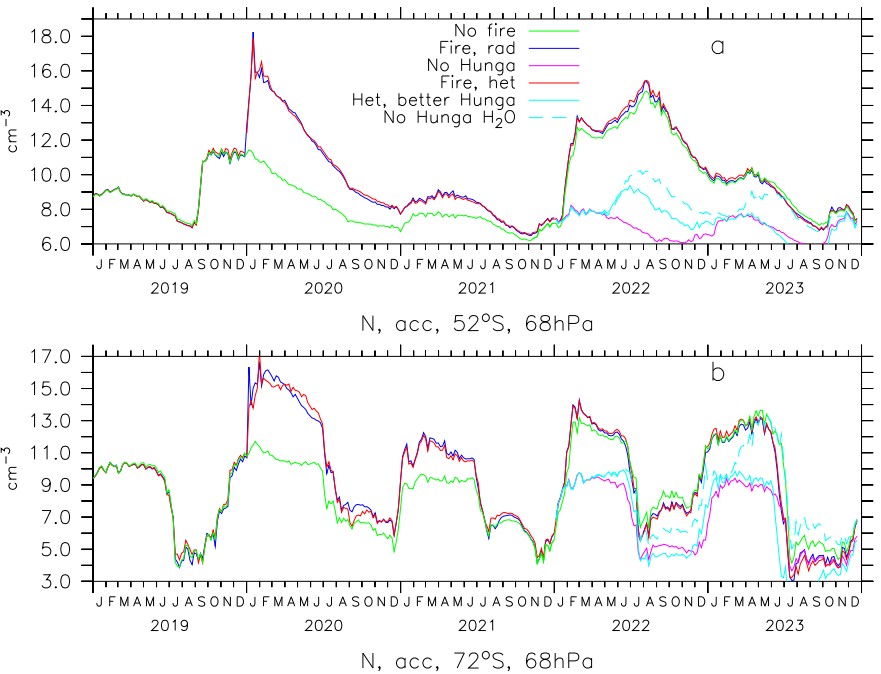

**Figure A4.** Calculated particle number density (accumulation mode, cm$^{-3}$), colors as in Table 2.

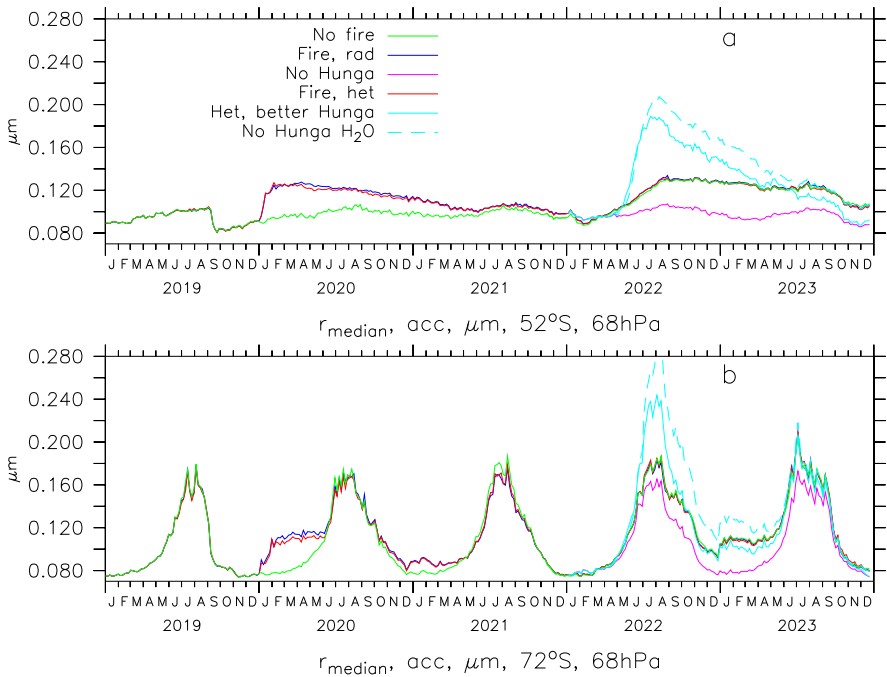

**Figure A5.** Calculated median wet particle radius (accumulation mode, $\mu$m), colors as in Table 2.

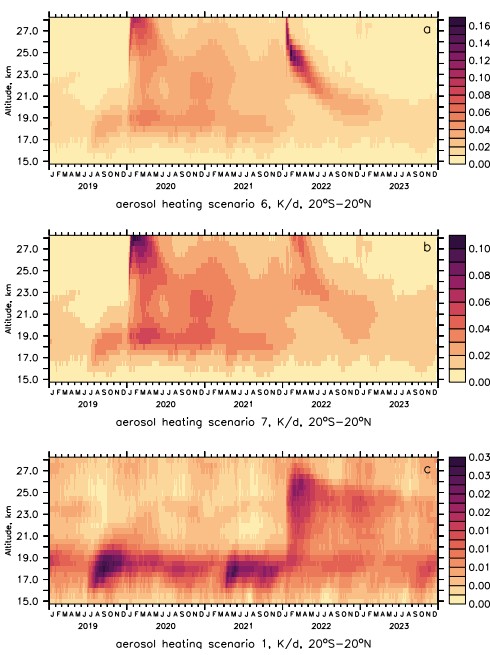

**Figure A6.** Calculated aerosol radiative heating rates (SW + LW) in tropics. a with fires (scenario 6), b same but no Hunga water (scenario 7), c without fires (scenario 1).

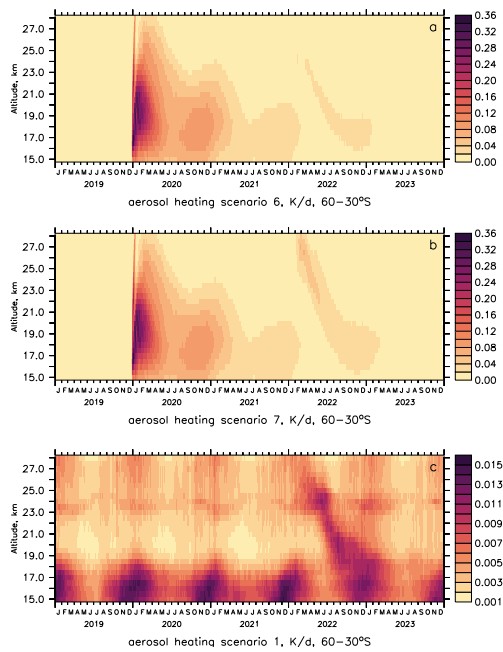

**Figure A7.** Calculated aerosol radiative heating rates in SH midlatitudes, panels as Fig. A6.

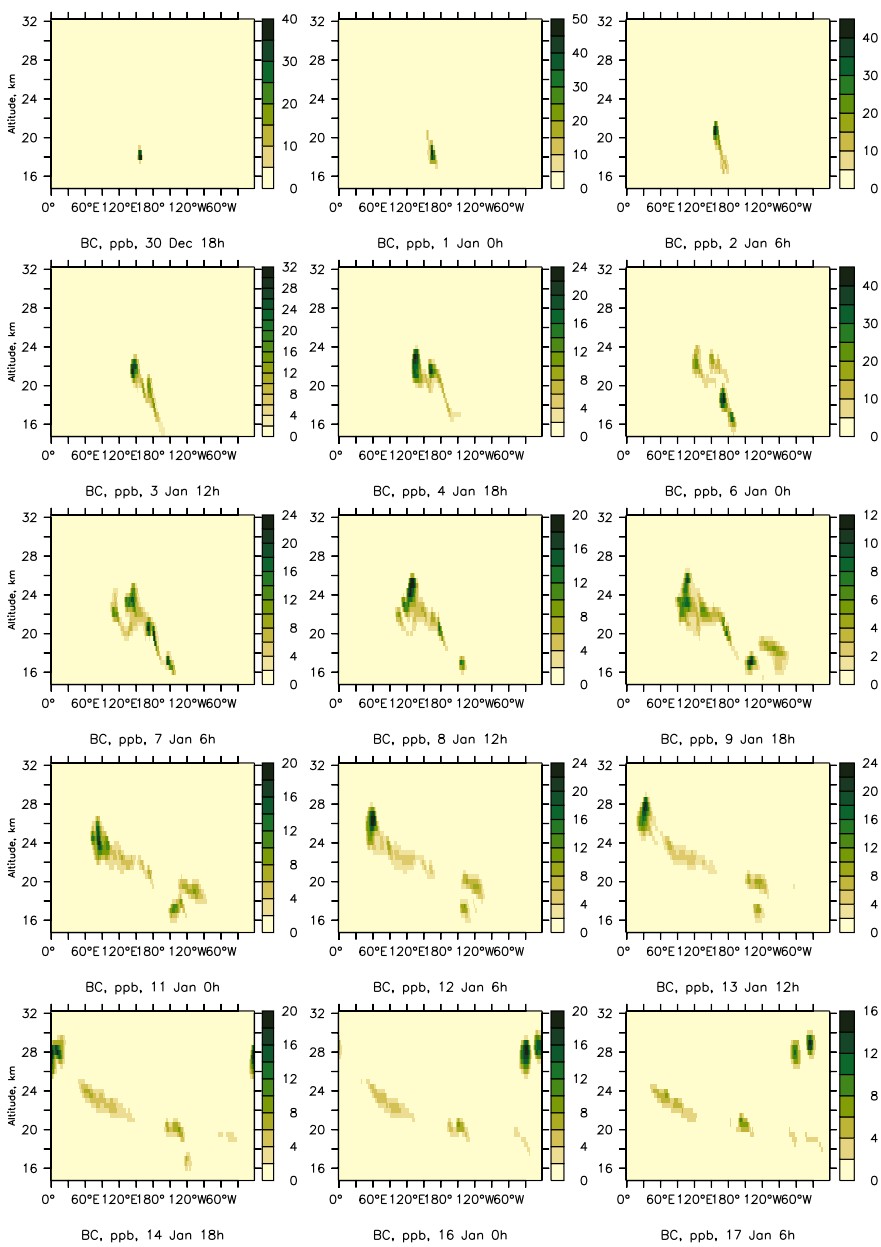

**Figure A8.** Simulated BC in lofting vortices at $35^oS$ against altitude in km, every 30 h, beginning on December 30, 2019.

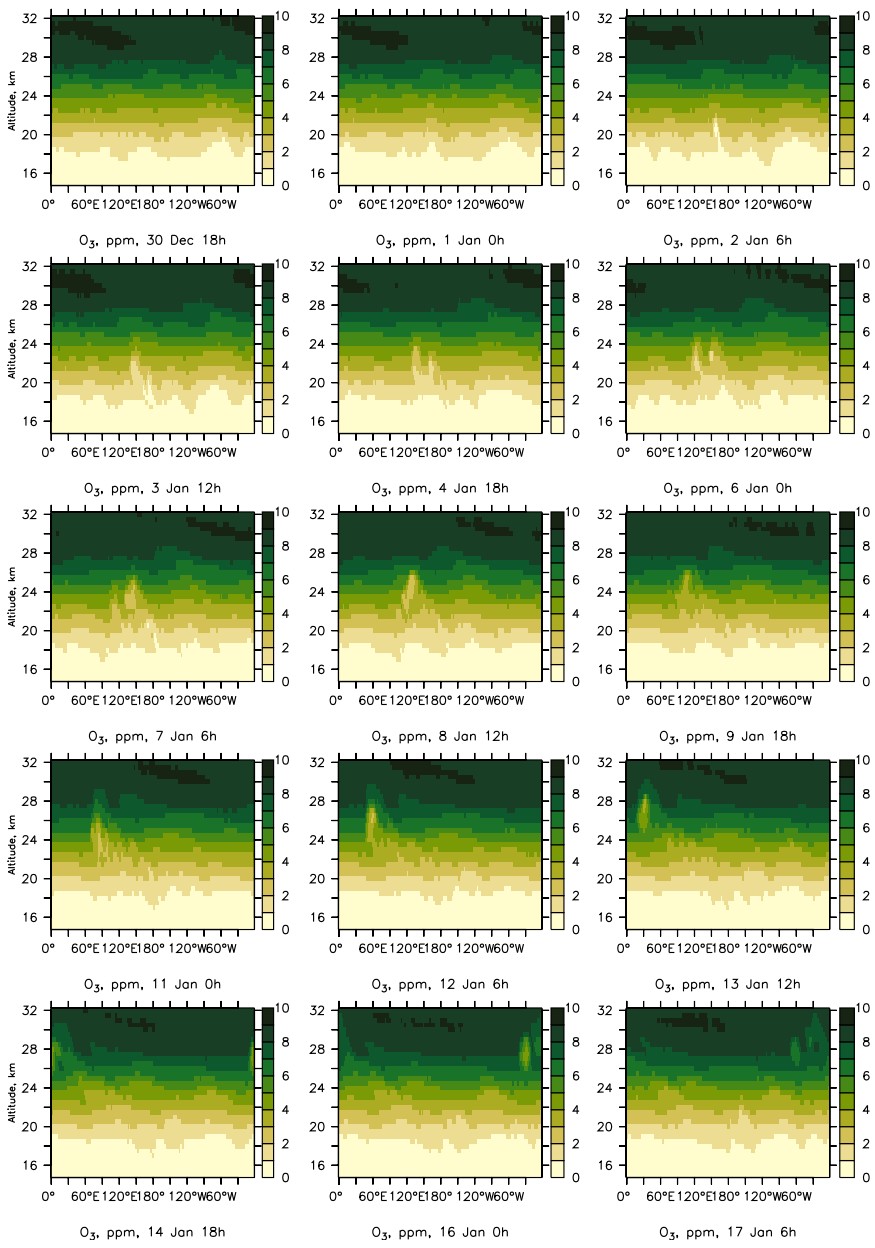

**Figure A9.** Simulated depleted ozone in the vortices of Fig. A8.

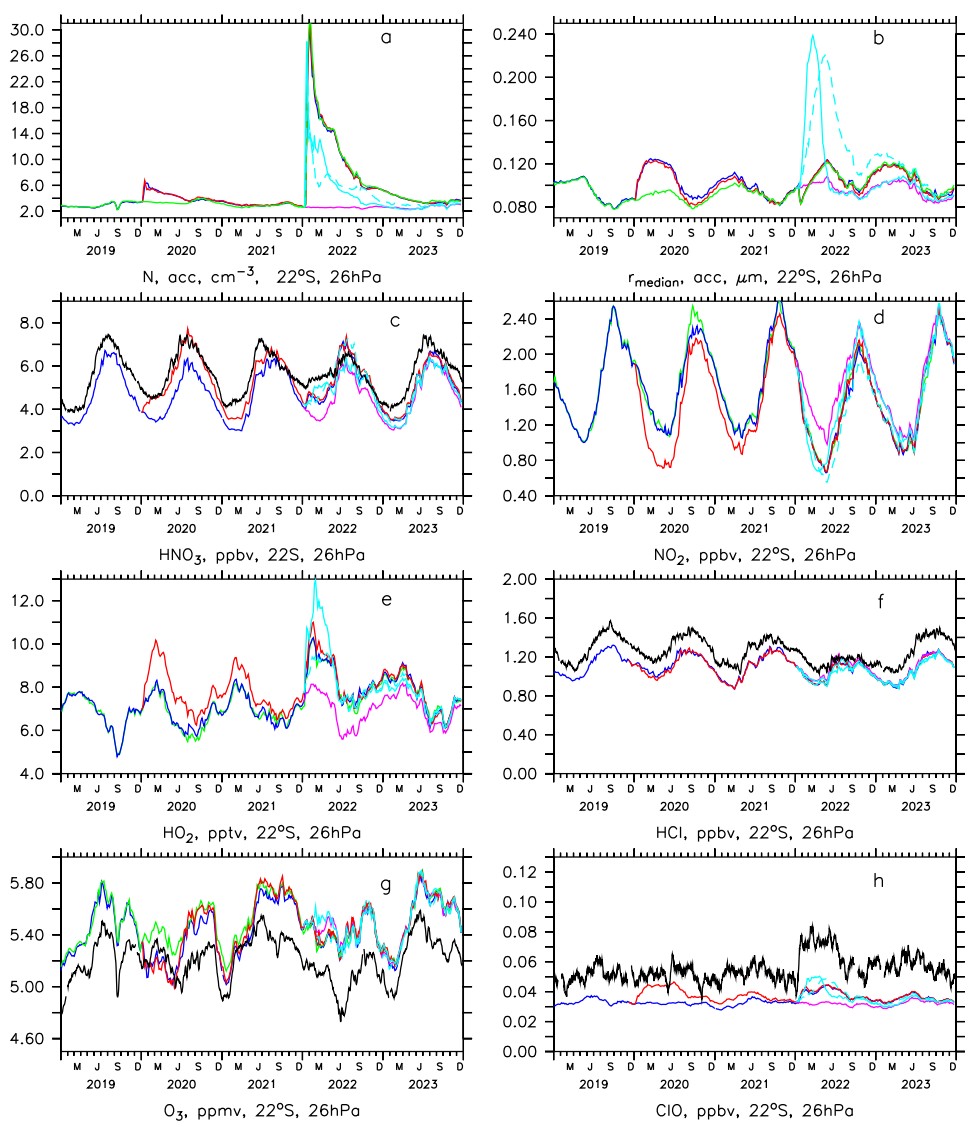

**Figure A10.** Number concentration, median radius (accumulation mode) and chemistry at latitude and altitude of the Hunga injection, colors see Table 2 and Fig. A1. MLS ClO observations are shown with 15d boxcar smoothing.