# Peer review of "Radiative forcing and stratospheric ozone changes due to major forest fires and recent volcanic eruptions including Hunga Tonga"

_EGUsphere, 2025_

## Referee Comment (RC2)

Review of manuscript 2025-2981 for Atmospheric Chemistry and Physics
"*Radiative forcing and stratospheric ozone changes due to major forest fires and recent volcanic eruptions including Hunga Tonga*", by Christoph Bruehl, Matthias Kohl & Jos Lelieveld

This manuscript presents findings from interactive stratospheric aerosol and chemistry simulations with a 3D composition climate model, to quantify the radiative and chemical effects on the stratosphere from recent intense wildfires and explosive volcanic eruptions.

The January 2022 Hunga eruption is found to cause a modest depletion of the stratospheric ozone layer throughout the 2022 Southern Hemisphere, with also the intense wildfires from Australia (Dec 2019) and Canada (NH summer 2019 and 2020) causing additional ozone loss in 2019 and 2020.

These impacts are found to be caused primarily by stratospheric aerosol particles from these sources, and although the emission of water vapour from Hunga is found to have altered the vertical location of the aerosol and caused some PSC-driven impacts, the direct effects from the emitted water vapour are smaller than these indirect effects.

The scientific writing style is generally quite good, and the findings presented are generally well-described and summarised. However, two of the initial Figures within the paper need to be revised, to give a clearer presentation of the stratospheric AOD and SAD enhancements after Hunga.

Specifically, Figure 1 gives the impression that the forest fires caused larger global stratospheric AOD enhancement, which is not the case.

The magnitude of the observed global stratospheric AOD in 2022 summer has been established (Khaykin et al., 2022; Legras et al., 2022;  Knepp et al., 2024; Joerimann et al., 2025) to have been substantially higher in 2022 than in 2020 and 2021 (indeed 2022 had the highest since the Pinatubo aerosol in 1992), and this can be seen clearly for example within Figure 10c of Khaykin et al. (2022).

There is a similar issue with Figure 3, with the 2019-2023 variation in SAD shown only at 1 altitude level (68hPa), this being in the lowermost stratosphere, the altitude at which the wildfires emitted, but ~5-10km below the altitude of the Hunga aerosol.

Figures 18 and 19 of Knepp et al. (2024) show clearly that the Hunga aerosol was present at a higher altitude, much deeper into the stratosphere (20-30km; ~30-50hPa), and the 68 hPa level selected for the SAD comparison (Figure 3 upper-panel and lower-panel).

The submitted manuscript then presents readers with an incorrect impression that the 2022 Hunga SAD-enhancement was lower than that from the 2019-2020 wildfires, when it's simply that the stronger Hunga aerosol was at higher altitude than the level shown.

These are the 2 "main minor revisions", that Figures 1 and 3 as currently drawn do not provide an adequate representation of the relative impacts on the stratosphere. Figure 3 shows only the impacts in the lowermost stratosphere (68 hPa), and Figure 1 gives stronger-looking lines to the more-uncertain limb-scatter instruments (OMPS-LP, OSIRIS) and weaker-looking dashed lines to the gold-standard SAGE-III aerosol measurements (these going off-the-scale on the plot).

The aerosol extinction derived from the OMPS-LP and OSIRIS limb-scatter instruments is known to be much more uncertain than SAGE-III, due primarily to having to make assumptions on particle size. In contrast, the SAGE-III instrument does not have to make an assumption about particle size, the solar occulation method measuring the extinction of sunlight directly.

In summary, Figure 1 needs to be re-drawn with a solid line used for the GloSSAC stratospheric AOD, this being the benchmark stratospheric AOD dataset (currently represented by a dotted line), and the current solid line used for OSIRIS changed to a dotted line. The y-axis range also needs to be increased accordingly (see revision M-MR1 below). And Figure 3 needs to show an integrated

I note however these are entirely presentational issue, and the required revisions remain at a minor level, and it will I expect be a relatively quick job to amend the Figures for this change.

Overall the paper presents a very interesting quantification of the relative magnitude of the effects, and the later Figures do compare like-for-like with the radiative forcing, albeit those based on this particular model's predictions.

And then I am recommending minor revisions overall, referring to these Figure changes as the only two "main minor revisions" (M-MR1, M-MR2), these listed below with the specific info on the modifications needed for these 2 Figures. There may need also to be some changes tothe associated interpretation, the Hunga impacts to be given more prominence than currently, the 2022 enhancement to the stratospheric AOD clearly substantially larger than that from the wildfires in 2019-2021.

Whilst the model predictions of the ozone layer and radiative impacts may indicate a stronger effect from the fires, this is a model result, and I am aware from a current Hunga aerosol intercomparison (Aquila et al., in prep.) that the EMAC predicted Hunga aerosol is weaker than in other models.

The manuscript can of course present the model findings, but the model being low-biased compared to the main Hunga enhancement in 2022 (compared to the benchmark SAGE-III/GloSSAC observations) needs to be noted as a significant caveat in the interpretation.

With the current administration of the US, the SAGE-III instrument may well be ended prematurely, and then model-observation comparisons in the future may have to rely more exclusively on the limb-scatter instruments. The large difference between the OSIRIS limb

scatter stratospheric AOD retrievals, and that from the benchmark GloSSAC/SAGE-III solar occultation, during the Hunga period, illustrates the importance of retaining SAGE-III.

Figure 3 needs to be re-drawn to show impacts comparing vertically-integrated stratospheric column-SAD, then inclusive of both impacts, and providing readers of the article with a balanced representation of stratospheric variations through the 2019-2023 period.

The 3$^{rd}$ main minor revision relates to the axis-label presentation of the Figures being quite poor, the y-axis values in Figure 1 given simply as log10 of the stratospheric AOD, and there being no y-axis labels for the reader to appreciate what quantity is shown there.

Again, these presentational issues are relatively easy to correct. And whilst this is somewhat disappointing that the authors should have submitted an article that seems not yet ready for expert peer review, I am aware of the APARC Hunga impacts report has a deadline of 31$^{st}$ July for papers to be citable, and may have affected the authors decision to submit prematurely.

It is also understandable, though somewhat regrettable, that a time-pressed Topical Editor (TE) was not able to flag these basic axis-label issues within an initial TE review.

Overall this is a solid paper, and the article is certainly within scope for ACP, and the findings presented are important, and will be of substantial interest to the community, once the revised manuscript can adequately represent the stratosphere's variation through this period.

The scientific writing is generally quite good, and although the writing style is quite brief, the main results are presented in a fair way, aside from the initial Figures 1 and 3.

Main minor revisions

M-MR-1) Re-plot Figure 1 with solid black line for GloSSAC (benchmark for stratospheric AOD)

As noted in the general comments above, please switch-round the dotted and solid lines given currently for the stratospheric AOD for GloSSAC and OSIRIS.

This is needed because the OSIRIS is a limb-scatter instrument, then much less certain aerosol retrieval than for the SAGE-III instrument, providing the main basis of GloSSAC.

The colour black should also be reserved for observations, and please also use then change the model lines to be coloured green, with black used for the observation datasets.

Please also update the y-axis range, to include the peak post-Hunga GloSSAC sAOD values. The GloSSAC observed stratospheric AOD "off the scale" during 2022, in the lower-panel of the current version of Figure 1 (for southern hemisphere stratospheric AOD).

M-MR-2) The current Figure 3 needs to be re-drawn for vertically-integrated SAD, to include the main Hunga impacts being at higher altitude than those from the 2019/2020 wildfires.

This is particularly evident within Figure 3, the Surface Area Density shown only for 1 selected altitude (68hPa). It is clear when referring to Figures 18 and 19 from Knepp et al. (2024) or

Figure 7a from Joerimann et al. (2025) that the observed post-Hunga SAD is significantly higher in magnitude than the wildfires (and also at higher altitude).

Had a 50hPa or 30hPa level been chosen for this southern mid-latitude comparison (the upper panel of Figure 3) the reader would have been presented with a timeline showing only a very minor enhancement in 2020 and 2021, and a much larger enhancement in 2022 (see Knepp et al., 2024, Figure 19).

Other minor revisions

O-MR-1) Abstract, line 1 – It's important to be clear the causality requires a conditional clause. Please insert "can" before "affect stratospheric chemistry" (it's only the exceptionally intense wildfires in 2019/2020 that have been shown to have this effect).

O-MR-2) Abstract, line 3 – Similarly to above, the Abstract needs to be clear it is these particular wildfires that are shown to have this effect (the majority of wildfires do not have this effect).   Change "emitted from wildfires" to "emitted from intense wildfires in 2019/2020".

O-MR-3) Abstract, line 4 – Change "due to increased solubility of HCl in particles containing organic acids"  -- the "due to" is not quite right there – it's the abundance of the smoke in the stratosphere that causes the effect (from the intense fires).  Suggested re-wording is "related to a high solubility of HCl in aerosol particles containing organic acids".

O-MR-4) Abstract, line 5 – Change "the upward transport of the pollution plumes resulted in".

The "upward transport" could be misunderstood by some readers to mean a longer timescale effect. But it was actually the extreme intensity of the fire, that  then caused a very deep plume, to then detrain the smoke at altitudes above the tropopause.

Suggested re-wording is to change

"Following the 2019/2020 Australian megfires, the upward transport of the pollution plumes resulted in ozone depletion in the…" to

"The extreme intensity of the 2019/2020 Australian wildfires, meant the smoke plume reached the stratosphere, causing in significant ozone depletion in the…"

O-MR-5) Abstract, line 6 – Please change "It diminished column ozone" – the preceding sentence already stated there was significant ozone depletion, and this "It diminished" phrasing also needs to be improved for peer-reviewed article.

Suggest to change

"It diminished column ozone in the following two years…"

to

"With the long residence time in the stratosphere, the wildfire smoke diminished column ozone for the following two years….

O-MR-6) Introduction, line 16 – Re-word "Since about 2017 it was observed…".  This is the 1st sentence of the Introduction, and the "about" is really quite poor wording there.   Suggest to adapt the sentence, and cite Mike Fromm's earlier papers (e.g. Fromm et al., 2004, 2005) which pointed out these effects from intense boreal forest fires.

O-MR-7) Introduction, lines 19-20 – "This holds for the Australian wildfires" – not clear what was meant here.

O-MR-8) Introduction, line 24 – Re-word sentence – improve "like for example Hunga Tonga" for scientific language.

O-MR-9) Introduction, line 25 – Similarly, the tense here is not quite right, change "lifting" to "lifted".

O-MR-10) Introduction, line 28 – Again, improve the scientific writing here, re-word "This and the about 10km thick layer" to be more precise.  This is also the first time the depth of the smoke aerosol is mentioned, and suggest to hint at the timescale being significant prior to becoming this 10km deep, e.g. maybe "The initial layer of smoke aerosol progressed to a 10km depth enhancement ….." or similar.

O-MR-11) Introduction, line 30 – Scientific writing poor here also – "has been analyzed in several studies" – please be clear what type of studies these are – presumably it was meant the observational studies in the initial period after the eruption?  If you meant modelling studies, be clear if you mean interactive stratospheric aerosol studies -- and there's actually been relatively few of these – the study of Zhu et al. (2022) and Li et al. (2024)

O-MR-12) Line 40, heading for section 2 – the "and used satellite data" is not great.  And suggest to include the words "interactive aerosol" to be clear these are interactive stratospheric aerosol CCM simulations.   Suggest to change to:

"Interactive chemistry-climate model simulations and satellite measurement datasets"

O-MR-13) Section 2.1 -- Line 42 – It seems too abrupt to simply begin "EMAC consists of" – and I think the acronym has not yet been introduced.  Suggest to re-word to:

"For this study, we analyse interactive stratospheric aerosol simulations with the ECHAM/MESSY Atmospheric Chemistry model (EMAC)…".

O-MR-14) Line 44 – put "up to 100 hPa" in brackets here, i.e. "(up to 100 hPa)".

O-MR-15) Line 54 – Re-word "accelerating the classical PCS reactions for chlorine activation also in mid-latitudes".  I think you meant that the presence of the extra aerosol means there's substantial additional heterogeneous chemistry occurring in mid-latitude, adding to that on the PSCs at high latitudes.  Please re-word to better explain what you meant here.

O-MR-16) Line 80 – Section 2.2 heading – change "OSIRIS" to "OSIRIS aerosol extinction".

O-MR-17) Line 89 – Section 2.3 heading – change "OMPS-LP" to "OMPS-LP aerosol extinction".

O-MR-18) Line 99 – Section 2.4 heading: change "AURA-MLS" to "Aura-MLS trace gas retrievals"

O-MR-19) Lines 105-107 – Please specify which species you are using to evaluate the model here.

O-MR-20) Caption to Figures 1, 2 and 3– The captions to these Figures notes 500 kt SO2 in the red-line sensitivity simulation, but this is difficult for a time-pressed reader to interpret.   Please add "All simulations other than the no-Hunga or 500 kt Hunga run apply 400 kt of SO2.

**References**

Fromm et al. (2004) "New directions: Eruptive transport to the stratosphere: Add fire-convection to volcanoes", *Atmos. Environ.*, vol. 38, 163-165.

Fromm et al. (2005) "Pyro-cumulus injection of smoke to the stratosphere: Observations and impact of a super blow-up in northwestern Canada on 3-4 August 1998", J. Geophys. Res., 110, https://doi.org/10.1029/2004JD005350

Joerimann, A., Sukhodolov, T., Luo, B. et al. (2025), "A REtrieval Method for optical and physical Aerosol Properties in the stratosphere (REMAPv1)",  in review, in Geosci. Mod. Dev., https://doi.org/10.5194/egusphere-2025-145

Khaykin, S., Podklajen, A., Ploeger, F. et al. (2022), "Global perturbation of stratospheric water and aerosol burden by Hunga eruption" , Comm. Earth Env., https://doi.org/10.1038/s43247-022-00652-x

Knepp, T., Kovilakam, M., Thomason, L. et al. (2024), "Characterization of stratospheric particle size distribution uncertainties using SAGE II and SAGE III/ISS extinction spectra", *Atmos. Meas. Tech.*, 17, 2025–2054, 2024, https://doi.org/10.5194/amt-17-2025-2024

Li, C., Peng, Y, E. Asher et al. (2024) "Microphysical simulations of the 2022 Hunga volcano eruption using a sectional aerosol model", Geophys. Res. Lett., vol. 51, issue 11., https://doi.org/10.1029/2024GL108522

Zhu, Y., Bardeen, C., Tilmes, S. et al. (2022) "Perturbations in stratospheric aerosol evolution due to the water-rich plume of the 2022 Hunga-Tonga eruption", Comms. Earth Env., 3, 248 https://doi.org/10.1038/s43247-022-00580-w

---

## Author Comment (AC1)

**Short reply to major remarks on "Radiative forcing and stratospheric ozone changes due to major forest fires and recent volcanic eruptions including Hunga Tonga"**

**1 General**

We thank the reviewers for their constructive remarks on improvement of figures and text, including references that we were not aware of that they should be discussed in the paper. We reply here to the major points and misunderstandings concerning the main focus of the paper, a detailed point by point reply will follow in the final response phase. The reviews and discussion in the APARC Hunga Tonga community convinced us to perform another sensitivity study with more realistic Hunga water vapor which will be included in the revised paper. In the following the reviewer's remarks are in bold.

**2 Review 1**

**However, the paper suffers from a lack of an appropriate literature review, especially regarding science around Hunga and its effect on ozone loss. The authors also need to discuss how their work falls within the existing literature within the discussion of their results.**
We will expand the introduction and the discussion on that, especially with regard to Hunga Tonga water vapor. This will include a statement that our main focus is heterogeneous chemistry in the lower stratosphere and on radiation.
**The author's need to do a proper literature review regarding Hunga in particular. I know you mention Santee et al., (2024) and direct readers there for other studies, but you are modelling Hunga impacts, so please provide a proper literature review here and later on discuss your results in the context of these references and their conclusions.**
We selected the references regarding our main focus, but agree with the referee that some additional material on gas phase effects of Hunga should be included in the discussion.
**I find the description of the simulations in section 2.1 and 3 very confusing. You mention, in multiple instances, simulations where you co-inject SO2. A lot of times I interpreted the co-injection as a sensitivity simulation, but how does this differ from what actually happened? For example on line 138 you mention an experiment with co-injection. Is the only difference between this run and the normal Hunga run an extra 100 Kt of SO2? Was the normal hunga run also have co-injected H2O and SO2? These sections need to be made clearer, with each distinct simulation having a title. It would make things even clearer if the titles of the different simulations were included in a legend in the figures. A table that shows the different**

**simulations with what is and isn't included would also be welcome.**
This will be improved. A table at the beginning of the results-section will help.
'Co-injection' with respect to the vertical distribution of $SO_2$ and $H_2O$ from
Hunga was considered only in one scenario, in the others the Hunga $SO_2$ injection was estimated from 3D OSIRIS observations covering a latitude belt and
Hunga $H_2O$ was injected in a relatively small slab (line 63ff) leading to too
much loss by ice formation. The table and the figures will include a more realistic Hunga scenario where $SO_2$ and $H_2O$ are injected in similar spatial patterns
with the $H_2O$ injection occuring about 1.5 km higher than the $SO_2$ one to avoid
most of the ice formation.

**For example, Figure 4 can be a 4 panel figure separating HCl and
ClO. That way readers could actually distinguish ClO. Figures also
need axis labels and panel letters.**
We agree, see the revised Fig. 4 of preprint with colors changed according to referee 2 (Fig.1). The missing panel letters were due to a problem in the graphics
software 'ferret'.

**A major concern I have is your model's timing of August HCl
recovery in the polar region in 2020-2023. In 2019, HCl in your
model control simulation recovers similarly to MLS, but in 2020, HCl
in your control and experiments seem to recover almost a month
later than MLS. This will give an extra month (September) where
you will likely have enhanced activated chlorine due to enhanced het.
chem than what is likely occurring in MLS and therefore you may be
overestimating your ozone loss. If HCl in your experiments recover
later than your control it would be different, but they are recovering
at the same time, and the timing does not agree with MLS. This
needs to be addressed by comparing with MLS HCl climatology and
investigated if it is a dynamical or chemical issue in your simulations.**
The time of the HCl recovery depends on size and depth of the ozone hole
(Grooß et al., 2011). If local ozone is close to zero HCl recovers quickly, if some
is left, e.g. due to advection, recovery takes longer (see Fig. 7 of preprint).
Model uncertainties on this are largest near the vortex edge.

**Line 123-124: The author's state "In April 2023 there appears to
be a southern midlatitude volcanic event missing in our inventory
or the Hunga Tonga SO2 injection is underestimated." What do you
mean there appears to be? If you think there is an event missing
in your dataset, please check it, and then either correct it or definitively state that it is missing. Also, why would an underestimation
of the Hunga SO2 injection affect April 2023 values, a year after the
eruption, but not earlier?**
I downloaded the newest version of GloSSAC where this feature is not present
in contrast to OSIRIS and OMPS-LP (Fig.2). Text will be modified.

**Line 197-198: "Hunga Tonga water vapour had only a small effect
on ozone and radiative forcing." Is this discussed anywhere in the
paper? I can't find it, but apologies if I missed it. If it is not discussed please add it in and also please compare to existing literature.**

**For example: Zhang et al., 2024 shows ozone decreases in the mid-latitudes due to the diluting of aerosols increasing the HOBr +HCl reaction. Wilmouth et al., 2023 also shows gas phase ozone loss in the midlatitudes due to elevated OH from the injected water vapor? How do your results compare with these and other papers?**

Text will be expanded using the simulation with more realistic Hunga water vapor. This will include short comparisons with Wilmouth et al. (2023) and Zhang et al. (2024). Some results, e.g. for $22^oS$ and 26hPa, might be shown in the Appendix.

**3 Review 2**

**Specifically, Figure 1 gives the impression that the forest fires caused larger global stratospheric AOD enhancement, which is not the case. The magnitude of the observed global stratospheric AOD in 2022 summer has been established (Khaykin et al., 2022; Legras et al., 2022; Knepp et al., 2024; Joerimann et al., 2025) to have been substantially higher in 2022 than in 2020 and 2021 (indeed 2022 had the highest since the Pinatubo aerosol in 1992), and this can be seen clearly for example within Figure 10c of Khaykin et al. (2022).**

The underestimate of Hunga SAOD in EMAC was an artifact of the used $SO_2$ injection method based on OSIRIS. In the new simulation SAOD enhancement by Hunga is much larger in tropics and southern midlatitudes and more consistent with the most recent GloSSAC data, see Fig.2, light blue curve. Text will be adjusted and expanded.

**There is a similar issue with Figure 3, with the 2019-2023 variation in SAD shown only at 1 altitude level (68hPa), this being in the lowermost stratosphere, the altitude at which the wildfires emitted, but 5-10km below the altitude of the Hunga aerosol.**

The main focus of this paper is on the lower stratosphere but a panel on the tropical middle stratosphere is included in the revised version to avoid confusion, see Fig. 3.

**In summary, Figure 1 needs to be re-drawn with a solid line used for the GloSSAC stratospheric AOD, this being the benchmark stratospheric AOD dataset (currently represented by a dotted line), and the current solid line used for OSIRIS changed to a dotted line. The y-axis range also needs to be increased accordingly (see revision M-MR1 below).**

Has been done, see Fig.2. The color scheme has been adjusted in all figures, using a full black line for observations.

**Figure 3 needs to be re-drawn to show impacts comparing vertically-integrated stratospheric column-SAD, then inclusive of both impacts, and providing readers of the article with a balanced representation of stratospheric variations through the 2019-2023 period.**

Such an integrated quantity does not represent the strongly temperature

and composition dependent heterogeneous chemistry where local SAD is for. Integrated quantities are shown in Fig. 1 (or Fig.2 here) but useful only for radiation effects.

**Again, these presentational issues are relatively easy to correct. And whilst this is somewhat disappointing that the authors should have submitted an article that seems not yet ready for expert peer review, I am aware of the APARC Hunga impacts report has a deadline of 31st July for papers to be citable, and may have affected the authors decision to submit prematurely.**

This is exactly the case and led to the improved scenario mentioned earlier.

**References**

Grooß, J.-U., Brautzsch, K., Pommrich, R., Solomon, S., and Müller, R.: Stratospheric ozone chemistry in the Antarctic: what determines the lowest ozone values reached and their recovery?, Atmos. Chem. Phys., https://doi.org/10.5194/acp-11-12217-2011, 2011.

Wilmouth, D. M., Østerstrøm, F. F., Smitha, J. B., Anderson, J. G., and Salawitch, R. J.: Impact of the Hunga Tonga volcanic eruption on stratospheric composition, PNAS, 120, https://doi.org/10.1073/pnas.2301994120, 2023.

Zhang, J., Wang, P., Kinnison, D., Solomon, S., Guan, J., Stone, K., and Zhu, Y.: Stratospheric Chlorine Processing After the Unprecedented Hunga Tonga Eruption, Geophys. Res. Lett., https://doi.org/10.1029/2024GL108649, 2024.

[Figure]

Figure 1: Simulated and observed HCl at 68 hPa at 52°S (a) and 72°S (c). Blue: EMAC with Hunga Tonga including water and dynamical effects of fires, Purple: EMAC without Hunga Tonga, Red: EMAC with Hunga Tonga and enhanced heterogeneous chlorine activation on organic particles. Black: MLS observations. Green curve for EMAC without fires not shown because very close to the blue one. Sensitivity studies close to the red curve not shown. Dashed curves for calculated ClO (b and d, MLS not shown). 5 day averages for EMAC, daily averages for MLS.

[Figure]

Figure 2: Stratospheric aerosol optical depth in tropical latitudes (b), northern (a) and southern midlatitudes (c), zonal 5 day averages. Green: EMAC without fires, blue: EMAC with 3 major fires (w/o heterogeneous chemistry on organic aerosol), red: EMAC with heterogeneous chemistry on smoke particles, red dashed: same with 500 kt of $SO_2$ from Hunga Tonga, light blue: same but $SO_2$ almost co-injected with $H_2O$ but $H_2O$ injection shifted 1.5km upward, purple: EMAC without Hunga Tonga, black dashed: observed by OSIRIS, black: GloSSAC, black dotted: observed by OMPS-LP.

[Figure]

Figure 3: Calculated surface area density $[\mu m^2/cm^3]$ in the middle stratosphere in southern subtropics (a) and the lower stratosphere in southern middle (b) and high (c) latitudes. Blue: sulfate and PSCs only with dynamical effect of smoke; Purple: without Hunga Tonga; Red: Including smoke particles; Red dashed and lightblue as in Fig.2

---

## Author Response (AR1)

**Detailed reply to the referee remarks on "Radiative forcing and stratospheric ozone changes due to major forest fires and recent volcanic eruptions including Hunga Tonga"**

**1  General**

We thank the reviewers for their constructive remarks on improvement of figures and text, including references that we were not aware of which should be discussed in the paper. We tried to accommodate every remark in the revised manuscript. The line numbers still refer to the preprint. The revised manuscript includes also some changes in numbers and conclusions related to improved simulations as a consequence of exchange with the APARC Hunga community and requests in the reviews. In the following reviewer remarks are in bold and answers in normal font. Some reviewer remarks where no answer is expected appear in a small bold font. For convenience an important figure of the revised manuscript is included.

**2  Review 1**

**Bruhl et al.: "Radiative forcing and stratospheric ozone changes due to major forest fires and recent volcanic eruptions including Hunga Tonga" Uses the EMAC model to investigate recent extreme stratospheric events. i.e., The 2019/2020 Australian wildfires and the Hunga volcanic event. Their results add to the growing body of literature by reproducing the wildfire HCl solubility theory and by simulating Hunga effects with a different model (reproducing results is valid and important contribution to the literature).**

**However, the paper suffers from a lack of an appropriate literature review, especially regarding science around Hunga and its effect on ozone loss. The authors also need to discuss how their work falls within the existing literature within the discussion of their results.**
More references are included in the introduction. The paragraph at line 36 begins now with "The main focus of this paper is on aerosol and heterogeneous chemistry in the lower stratosphere which has the largest impact on radiative forcing and total ozone. For the analyzed period 2019 to 2023 the Southern Hemisphere is of particular interest." Our results are now discussed in the light of the existing literature.

**For example, do their results agree with Solomon et al. (2023) that proposed enhanced HCl solubility in wildfire smoke? How do their Hunga ozone depletion agree/disagree with other studies on Hunga and ozone.**

Comparison of results for chlorine activation by wildfire aerosol and Hunga included in discussion of Fig. 4 and 5. Also for ozone related to Fig. 7.

**The papers description of simulations performed and how figure results are presented also needs to updated to make the results and conclusions of the paper easier to understand.**
A table summarizing the simulations is now included in section 2. The figures are improved.

**Overall, the paper fits ACP's scope and would be a good contribution to the literature, but it currently needs major revisions. Please see below for details.**

**2.1 Major comments**

**The author's need to do a proper literature review regarding Hunga in particular. I know you mention Santee et al., (2024) and direct readers there for other studies, but you are modelling Hunga impacts, so please provide a proper literature review here and later on discuss your results in the context of these references and their conclusions.**
We largely agree. More references are implemented in section 1 and 3. Hunga $H_2O$ gas phase effects in the middle stratosphere are not the main focus of this paper as indicated in the revised introduction. Nevertheless we added a paragraph on middle stratosphere effects of Hunga at the end of section 3, including a figure in the Appendix.

**Same needs to be done when discussing the wildfire results.**
See specific comments.

**Some other studies are linked below.**
**Wilmouth, D. M., Østerstrøm, F. F., Smith, J. B., Anderson, J. G., and Salawitch, R. J.: Impact of the Hunga Tonga volcanic eruption on stratospheric composition, Proc. Natl. Acad. Sci. U.S.A., 120, e2301994120, https://doi.org/10.1073/pnas.2301994120, 2023. An important study looking at gas phase changes due to Hunga.**
**Wohltmann, I., Santee, M. L., Manney, G. L., and Millán, L. F.: The Chemical Effect of Increased Water Vapor From the Hunga Tonga-Hunga Ha'apai Eruption on the Antarctic Ozone Hole, Geophysical Research Letters, 51, e2023GL106980, https://doi.org/10.1029/2023GL106980, 2024. A modelling study looking at 2023 and Hunga in the Antarctic.**
**Zhou, X., Dhomse, S. S., Feng, W., Mann, G., Heddell, S., Pumphrey, H., Kerridge, B. J., Latter, B., Siddans, R., Ventress, L., Querel, R., Smale, P., Asher, E., Hall, E. G., Bekki, S., and Chipperfield, M. P.: Antarctic Vortex Dehydration in 2023 as a Substantial Removal Pathway for Hunga Tonga-Hunga Ha'apai Water Vapor, Geophysical Research Letters, 51, https://doi.org/10.1029/2023gl107630, 2024. A study on the Antarctic being a dehydration pathway for Hunga H2O. Also models at Antarctic**

ozone changes due to Hunga.

Manney, G. L., Santee, M. L., Lambert, A., Millán, L. F., Minschwaner, K., Werner, F., Lawrence, Z. D., Read, W. G., Livesey, N. J., and Wang, T.: Siege in the Southern Stratosphere: Hunga Tonga-Hunga Ha'apai Water Vapor Excluded From the 2022 Antarctic Polar Vortex, Geophysical Research Letters, 50, e2023GL103855, https://doi.org/10.1029/2023GL103855, 2023. An observational study on how Hunga H2O was excluded from the 2022 vortex.

Schoeberl, M. R., Wang, Y., Ueyama, R., Taha, G., Jensen, E., and Yu, W.: Analysis and Impact of the Hunga Tonga-Hunga Ha'apai Stratospheric Water Vapor Plume, Geophysical Research Letters, 49, https://doi.org/10.1029/2022gl100248, 2022. Looks at the radiative impact.

Fleming, E. L., Newman, P. A., Liang, Q., and Oman, L. D.: Stratospheric Temperature and Ozone Impacts of the Hunga Tonga-Hunga Ha'apai Water Vapor Injection, JGR Atmospheres, 129, e2023JD039298, https://doi.org/10.1029/-2023JD039298, 2024. A modelling study that includes the Antarctic ozone response of Hunga.

Krzysztof Wargan, Gloria L Manney, Nathaniel J Livesey. Factors contributing to the unusually low Antarctic springtime ozone in 2020-2023. ESS Open Archive . February 12, 2025. DOI: 10.22541/essoar.173940272.21623654/v1 (preprint). This is a preprint but could give some good insights.

Regarding modelling HCl solubility, the authors are using the methodology of Solomon et al., (2023) and therefore need to make sure they mention that their results are agreeing/confirming (or not) with that study, but with a different model.
See answer to specific comments.

I find the description of the simulations in section 2.1 and 3 very confusing. You mention, in multiple instances, simulations where you co-inject SO2. A lot of times I interpreted the co-injection as a sensitivity simulation, but how does this differ from what actually happened? For example on line 138 you mention an experiment with co-injection. Is the only difference between this run and the normal Hunga run an extra 100 Kt of SO2? Was the normal hunga run also have co-injected H2O and SO2? These sections need to be made clearer, with each distinct simulation having a title. It would make things even clearer if the titles of the different simulations were included in a legend in the figures. A table that shows the different simulations with what is and isn't included would also be welcome.
A table with the most important features of the simulations is incorporated including the color code for the figures (see also short reply).

A lot of the results presented in the figures are hard to distinguish. A lot of times there is just too much going on in each figure and combined with a lack of legends can make it hard to process without having to continuously read the caption to understand what each line

**represents. I suggest including legends and expanding some of the figures into more panels. For example, Figure 4 can be a 4 panel figure separating HCl and ClO. That way readers could actually distinguish ClO. Figures also need axis labels and panel letters.**

HCl and ClO in Figure 4 are now in separate panels as suggested. Panel letters are included now in every figure. Labels or titles are at the bottom of each panel (Ferret-style).

**I find it hard to distinguish how large/significant the wildfire or Hunga induced anomalies from your experiments are compared to your control without also presenting an MLS baseline and variability.** Introducing detrending and/or longtime averages in observations and simulations always comes with additional uncertainties. Since we use lower boundary conditions from time dependent observations together with nudged meteorology we prefer to compare the timelines directly. For comparison with averages we refer to Stone et al. (2025).

**A major concern I have is your model's timing of August HCl recovery in the polar region in 2020-2023. In 2019, HCl in your model control simulation recovers similarly to MLS, but in 2020, HCl in your control and experiments seem to recover almost a month later than MLS. This will give an extra month (September) where you will likely have enhanced activated chlorine due to enhanced het. chem than what is likely occurring in MLS and therefore you may be overestimating your ozone loss. If HCl in your experiments recover later than your control it would be different, but they are recovering at the same time, and the timing does not agree with MLS. This needs to be addressed by comparing with MLS HCl climatology and investigated if it is a dynamical or chemical issue in your simulations.** The time of the HCl recovery depends on size and depth of the ozone hole (Grooß et al., 2011). If local ozone is close to zero HCl recovers quickly, if some is left, e.g. due to advection, recovery takes longer (see Fig. 7 of preprint). Model uncertainties on this are largest near the vortex edge. Text modified.

**2.2 Specific comments**

**Line 2: The author's state "Similar to major volcanic eruptions". Dynamically similar, sure. But, chemically it seems to be quite different.**
We cannot expand that here because of the word limit. 'heterogenous' added in introduction (second sentence).

**Line 2-5: Please change: "Using the chemistry climate model EMAC, we demonstrate that organic carbon emitted from forest fires, injected into the stratosphere through pyro-cumulonimbi, enhances**

heterogeneous chlorine activation due to increased solubility of HCl in particles containing organic acids and an augmented aerosol surface area". To "Using the chemistry climate model EMAC, we demonstrate that organic carbon emitted from forest fires, injected into the stratosphere through pyro-cumulonimbi, enhances heterogeneous chlorine activation due to increased solubility of HCl in particles containing organic acids and an augmented aerosol surface area, in agreement with existing literature." It is great that you are confirming previous results with a different model, but isn't a novel conclusion.
4 or 5 words might be added if some spaces are removed. Inserted as requested. I hope it works.

**Line 8 and other instances: I believe the current accepted naming is just "Hunga". Please check and update throughout the paper.**
*Really?* I have seen that yet only in the draft paper of Aquila et al. (2025) and in Santee et al. (2024). Nevertheless I introduced the short name in the introduction.

**Line 9: "The water vapour injection from the volcano altered only the vertical distribution of ozone loss." As opposed to what? Latitudinal? A restructuring of the vertical profile? I don't see presentation of these results anywhere apart from a mention in the conclusions.**
This is opposed to the sentence before. More details are presented in the results section.

**Line 17: "There are some other references you should consider mentioning here regarding dynamics" Yu et al. (2019); Damany-Pearce et al., (2023)**
Inserted.

**Yu, P., Toon, O. B., Bardeen, C. G., Zhu, Y., Rosenlof, K. H., Portmann, R. W., Thornberry, T. D., Gao, R.-S., Davis, S. M., Wolf, E. T., De Gouw, J., Peterson, D. A., Fromm, M. D., and Robock, A.: Black carbon lofts wildfire smoke high into the stratosphere to form a persistent plume, Science, 365, 587–590, https://doi.org/10.1126/science.aax1748, 2019.**
**Damany-Pearce, L., Johnson, B., Wells, A., Osborne, M., Allan, J., Belcher, C., Jones, A., and Haywood, J.: Australian wildfires cause the largest stratospheric warming since Pinatubo and extends the lifetime of the Antarctic ozone hole, Scientific Reports, 12, 1–15, https://doi.org/10.1038/s41598-022-15794-3, 2022.**
**Line 18: "More important references regarding wildfire chemistry" Stone et al., (2025); Solomon et al., (2022)**
Inserted.

**Stone, K., Solomon, S., Yu, P., Murphy, D. M., Kinnison, D., and Guan, J.: Two-years of stratospheric chemistry perturbations from the**

2019/2020 Australian wildfire smoke, Atmos. Chem. Phys., 25, 7683–7697, https://doi.org/10.5194/acp-25-7683-2025, 2025.

Solomon, S., Dube, K., Stone, K., Yu, P., Kinnison, D., Toon, O. B., Strahan, S. E., Rosenlof, K. H., Portmann, R., Davis, S., Randel, W., Bernath, P., Boone, C., Bardeen, C. G., Bourassa, A., Zawada, D., and De-genstein, D.: On the stratospheric chemistry of midlatitude wildfire smoke, Proceedings of the National Academy of Sciences of the United States of America, 119, 1–9, https://doi.org/10.1073/pnas.2117325119, 2022.

**Line 20: The authors state: "This holds for the Australian fires in December 2019 to January 2020 as well as Canadian fires in 2017" Are you referring to chemistry here? Because there is no direct evidence (only correlation) that 2017 and 2019 Northern Hemisphere wildfires affected chemistry. If you are referring to dynamics please adjust this sentence to be clearer.**
On a poster presented at EGU it was shown that these fires caused additional chlorine activation in the lower stratosphere of the Northern Hemisphere. Sentence added in the results section.

**Line 29: The author's are referencing a news article (Cutts). Please reference the actual papers. In this case it looks like Ansmann et al., 2023 and Solomon et al., 2023 that you have already reference elsewhere in the paper.**
Here it was intended to show that the subject has arrived in AGU's news. I suppose the reviewer means Ansmann et al. (2022) but this would be a repetition here. I added 2 references of Stone which fit better here.

**Line 46: What does "Slightly nudged mean?" Please define exactly how the QBO is nudged.**
Sentence expanded, now after link "using a relaxation time of 2 months to allow for a comparison with satellite data (Giorgetta and Bengsston, 1999)"

**Line 55. What are the reactions that are included in the model?**
A table on included heterogeneous reactions is added and referenced to in line 55.

**Line 60. If Kohl et al continues to 2023, when does the previous Shallock et al., dataset end?**
Now included in the next sentence: "from September 2019 to December 2023".

**Figure 1 caption: What is GloSSAC? Is it an observational dataset? A model? It isn't defined anywhere just mentioned again on lines 119 and 130**
Caption expanded. More in a new subsection of section 2. See Fig. 1.

**Line 123-124: The author's state "In April 2023 there appears to be a southern midlatitude volcanic event missing in our inventory**

or the Hunga Tonga SO2 injection is underestimated." What do you mean there appears to be? If you think there is an event missing in your dataset, please check it, and then either correct it or definitively state that it is missing. Also, why would an underestimation of the Hunga SO2 injection affect April 2023 values, a year after the eruption, but not earlier? Your model aerosols e-folding time could be incorrect too.

The text from line 123 to 130 is rewritten in the light of 2 new simulations and more in-sight into the GloSSAC data. There is indeed a too fast removal of too big particles near the source region in case of high water vapour.

**Line 124-125: The author's state: "Also in April 2023 some source of aerosol is missing in the tropics" I can't see this. It looks like you are overestimating SAD in the tropics compared to OSIRIS after April 2023. Please check.**

**Line 144-145: "Hunga Tonga causes a heating rate by up to 0.025 K/d at 25 km in the tropics and 0.011 K/d in southern midlatitudes." How does this compare to the current literature? Also in agreement with Yu et al. and other studies?**

These numbers were from the scenarios where extinction was underestimated. A further panel with the more realistic scenario is included in Figs A6 and A7 and the text expanded and updated

**Figure 5. Please be consistent in naming ClONO2. Figure 5 panel title uses ClNO3.**

Corrected.

**Line 165-167. Please discuss in context of existing literature (Solomon et al., 2023; Stone et al., 2025; Ma et al., 2024). If you are confirming results from Solomon et al., (2023) (but with a different model) please put that in your discussion.**

Inserted: "as also found by Solomon et al. (2023) and Stone et al. (2025) for both latitude regions in 2020. In 2021 this perturbation is larger than in Stone et al. (2025)." and "by up to 0.4 ppmv in agreement with Solomon et al. (2023)"

**Line 177-178: "and almost complete loss at the edge of the ozone hole in winter 2020 in agreement with MLS (black, red and green curves)." The ozone loss seen in the control run in the polar region looks comparable to MLS if you consider the offset in your model here. With this offset it is hard to see if your control run or your experiment better represent the observations.**

With the revised color scheme in figure 7 it is much better visible that ozone loss in scenario 4 or 6 is larger than in scenario 1 or 2 (almost 0.4ppmv, panel

b). Text modified.

**Line 180-184: Do you have simulations with combined dynamical and chemistry effects? Or are they separate? Again, it is hard to understand exactly what is happening in each simulation based on your descriptions.**
The combined effect you see when you subtract the results of scenario 1 from the ones of scenario 4 or 6. Clarified in text.

**Line 184: "The loss due to the eruption of Hunga Tonga is up to about 15 DU in 2022 and 2023 (Fig. 8b)." Where? Midlatitudes? Polar region? How does this compare to existing literature?**
"near the edge of the Antarctic vortex" inserted.

**Line 192-193: "To obtain agreement with the chlorine activation in polar regions observed by MLS it is essential to include the enhanced solubility of HCl in particles containing organic acids from major forest fires." Please add in a statement like "In agreement with Solomon et al., (2023)" or something similar.**
Inserted as suggested, thanks.

**Line 197-198: "Hunga Tonga water vapour had only a small effect on ozone and radiative forcing." Is this discussed anywhere in the paper? I can't find it, but apologies if I missed it. If it is not discussed please add it in and also please compare to existing literature. For example: Zhang et al., 2024 shows ozone decreases in the midlatitudes due to the diluting of aerosols increasing the HOBr +HCl reaction. Wilmouth et al., 2023 also shows gas phase ozone loss in the midlatitudes due to elevated OH from the injected water vapor? How do your results compare with these and other papers?**
Discussion of Fig. 4, 5 and 7 expanded and short sensitivity study on the HOBr effect mentioned.

**Line 199-: "Co-injection of SO2 and H2O leads to features not observed since it almost prevents the descent of the H2O plume by radiative cooling in the early phase due to heating of the sulfur aerosol which is formed too rapidly, related to strongly enhanced OH, which leads to a too late arrival of the additional water vapour in the lower stratosphere at mid and high southern latitudes." Again, I am confused my co-injected is presented like a sensitivity study. Is it not what actually happened? Or does co-injection mean at the same altitude?**
Text revised for clarification. In the new scenario in Table 2 SO2 and H2O are not exactly 'co-injected at the same gridpoints with the same vertical distribution.

**Line 204: "causes a change in the vertical distribution with almost no effect on total ozone," Where is this discussed/presented in the paper?**
Discussion now in last paragraph of Section 3.

**2.3 Technical corrections**

**Line 30. Change "Since decades" to "In decades"**
"in the last 3 decades".

**There are a lot of undefined acronyms throughout the paper. Please define acronym's when they first appear. For example, EMAC is not defined were it first appears on line 34. ECHAM5 on line 43, GMXE, EQSAM, MSBM etc.**
Mostly done but I don't like to explain every subroutine or module name which you find in Jöckel et al. (2010) and Giorgetta et al. (2006).

**Line 43: MESSy is spelled out, therefore please but in brackets.**
Done.

**Line 44: I believe ERA5 is typically not hyphenated.**
Corrected, but you see both in the literature.

**Line 54: "PSC"**
Corrected.

**Line 62. Suggest changing "Including several hundreds of explosive" to "Including several hundred explosive".**
Corrected (line 59!).

**Line 109-110: Please reword: "optical depth (SAOD) for tropics and southern and northern midlatitudes is" to "optical depth (SAOD) for the tropics, the southern midlatitudes, and the northern midlatitudes" or something similar.**
Corrected.

**Line 110-111: I believe you want a comma after this sentence instead of a period"1. Considering the organic aerosol from major forest fires is essential for agreement with the observations by OSIRIS."**
? OMPS inserted.

**Line 174. Do you mean H2O or are you referring to Figure A3?**
"$HO_2$ (Fig. A3)".

**Line 195: is it 28 DU or 30 DU as mentioned earlier?**
Corrected.

**Line 197: "up to 15 DU..." where is this occurring? Midlatitudes? Polar region?**
Inserted "near 70$^o$S".

**3   Review 2**

**This manuscript presents findings from interactive stratospheric aerosol and chemistry simulations with a 3D composition climate model, to quantify the radiative and chemical effects on the stratosphere from recent intense wildfires and explosive volcanic eruptions.**

**The January 2022 Hunga eruption is found to cause a modest depletion of the stratospheric ozone layer throughout the 2022 Southern Hemisphere, with also the intense wildfires from Australia (Dec 2019) and Canada (NH summer 2019 and 2020) causing additional ozone loss in 2019 and 2020.**

**These impacts are found to be caused primarily by stratospheric aerosol particles from these sources, and although the emission of water vapour from Hunga is found to have altered the vertical location of the aerosol and caused some PSC-driven impacts, the direct effects from the emitted water vapour are smaller than these indirect effects.**

**The scientific writing style is generally quite good, and the findings presented are generally well-described and summarised. However, two of the initial Figures within the paper need to be revised, to give a clearer presentation of the stratospheric AOD and SAD enhancements after Hunga.**
**Specifically, Figure 1 gives the impression that the forest fires caused larger global stratospheric AOD enhancement, which is not the case. The magnitude of the observed global stratospheric AOD in 2022 summer has been established (Khaykin et al., 2022; Legras et al., 2022; Knepp et al., 2024; Joerimann et al., 2025) to have been substantially higher in 2022 than in 2020 and 2021 (indeed 2022 had the highest since the Pinatubo aerosol in 1992), and this can be seen clearly for example within Figure 10c of Khaykin et al. (2022).**
The underestimate of Hunga SAOD in EMAC was an artifact of the used SO$_2$ injection method based on OSIRIS. In the new simulation SAOD enhancement by Hunga is much larger in tropics and southern midlatitudes and more consistent with the most recent GloSSAC/SAGEIII data, see Fig.1, light blue and thick black curves.

**There is a similar issue with Figure 3, with the 2019-2023 variation in SAD shown only at 1 altitude level (68hPa), this being in the lowermost stratosphere, the altitude at which the wildfires emitted,**

**but 5-10km below the altitude of the Hunga aerosol. Figures 18 and 19 of Knepp et al. (2024) show clearly that the Hunga aerosol was present at a higher altitude, much deeper into the stratosphere (20-30km; 30-50hPa), and the 68 hPa level selected for the SAD comparison (Figure 3 upper-panel and lower-panel).**
As indicated in the public short reply and the introduction, our main focus is on the lower stratosphere. Nevertheless, we include now a panel in this figure for the altitude and latitude where Hunga had its largest effect. Fig.3 shows SAD seen by heterogeneous chemistry in our model which is not identical with the one derived for radiation or from satellite data.

**The submitted manuscript then presents readers with an incorrect impression that the 2022 Hunga SAD-enhancement was lower than that from the 2019-2020 wildfires, when it's simply that the stronger Hunga aerosol was at higher altitude than the level shown.**
It should be clear now that this depends on altitude and what our focus is.

**These are the 2 "main minor revisions", that Figures 1 and 3 as currently drawn do not provide an adequate representation of the relative impacts on the stratosphere. Figure 3 shows only the impacts in the lowermost stratosphere (68 hPa), and Figure 1 gives stronger-looking lines to the more- uncertain limb-scatter instruments (OMPS-LP, OSIRIS) and weaker-looking dashed lines to the gold-standard SAGE-III aerosol measurements (these going off-the-scale on the plot).**
Figure 1 now contains SAGEIII ISS cloud cleared from GloSSAC as thick black line, the other instruments are dashed or dotted, model simulations are in colors indicated in Table 2 as suggested by reviewer 1 (see included figure).

**The aerosol extinction derived from the OMPS-LP and OSIRIS limb-scatter instruments is known to be much more uncertain than SAGE-III, due primarily to having to make assumptions on particle size. In contrast, the SAGE-III instrument does not have to make an assumption about particle size, the solar occulation method measuring the extinction of sunlight directly.**
Yes, I mentioned now in the text that there can be artifacts. However, even SAGEIII can have cloud contamination, seen in the unfiltered SAGE data of GlosSSAC shown dash-dotted in Fig.1 to get some feeling for the uncertainty. Text modified.

**In summary, Figure 1 needs to be re-drawn with a solid line used for the GloSSAC stratospheric AOD, this being the benchmark stratospheric AOD dataset (currently represented by a dotted line), and the current solid line used for OSIRIS changed to a dotted line. The y-axis range also needs to be increased accordingly (see revision M-MR1 below).**
Done.

I note however these are entirely presentational issue, and the required revisions remain at a minor level, and it will I expect be a relatively quick job to amend the Figures for this change.

Overall the paper presents a very interesting quantification of the relative magnitude of the effects, and the later Figures do compare like-for-like with the radiative forcing, albeit those based on this particular model's predictions.

And then I am recommending minor revisions overall, referring to these Figure changes as the only two "main minor revisions" (M-MR1, M-MR2), these listed below with the specific info on the modifications needed for these 2 Figures. There may need also to be some changes to the associated interpretation, the Hunga impacts to be given more prominence than currently, the 2022 enhancement to the stratospheric AOD clearly substantially larger than that from the wildfires in 2019-2021.

Nevertheless, the impact of the Australian wildfires on total ozone is larger than the one of Hunga (models and observations). The same holds for global instantaneous radiative forcing, at least in the first year after the perturbation.

Whilst the model predictions of the ozone layer and radiative impacts may indicate a stronger effect from the fires, this is a model result, and I am aware from a current Hunga aerosol intercomparison (Aquila et al., in prep.) that the EMAC predicted Hunga aerosol is weaker than in other models.

The manuscript can of course present the model findings, but the model being low-biased compared to the main Hunga enhancement in 2022 (compared to the benchmark SAGE-III/GloSSAC observations) needs to be noted as a significant caveat in the interpretation.

Yes, the presented simulations in the first version of the manuscript indeed underestimated aerosol and water vapour perturbations by Hunga. I'm in close contact to V. Aquila on that and in the actual version a strongly improved scenario for Hunga is included which is also available for the APARC Hunga model intercomparison community. The problem is that results are extremely sensitive to the spatial distribution of the injections of Hunga $SO_2$ and water vapour. This is now included in the conclusions.

With the current administration of the US, the SAGE-III instrument may well be ended prematurely, and then model-observation comparisons in the future may have to rely more exclusively on the limb-scatter instruments. The large difference between the OSIRIS limb scatter stratospheric AOD retrievals, and that from the benchmark GloSSAC/SAGE-III solar occultation, during the Hunga period, illustrates the importance of retaining SAGE-III.

**Figure 3 needs to be re-drawn to show impacts comparing vertically-integrated stratospheric column-SAD, then inclusive of both impacts, and providing readers of the article with a balanced representation of stratospheric variations through the 2019-2023 period.**
This would be misleading here since such an integrated quantity does not represent the strongly temperature and composition dependent heterogeneous chemistry where local SAD in our definition indicated in the caption of Fig.3 is for. In the revised version Hunga dominates SAD in the southern subtropics at 26hPa while at 68hPa Hunga and wildfire effects on SAD are similar. Integrated quantities are shown in Fig.1 but useful only for radiation effects.

**The 3rd main minor revision relates to the axis-label presentation of the Figures being quite poor, the y-axis values in Figure 1 given simply as log10 of the stratospheric AOD, and there being no y-axis labels for the reader to appreciate what quantity is shown there.**
Fig. 1 is now linear (see above).

**Again, these presentational issues are relatively easy to correct. And whilst this is somewhat disappointing that the authors should have submitted an article that seems not yet ready for expert peer review, I am aware of the APARC Hunga impacts report has a deadline of 31st July for papers to be citable, and may have affected the authors decision to submit prematurely.**
Yes, this was the case, but the discussion lead to improved scenarios.

**It is also understandable, though somewhat regrettable, that a time-pressed Topical Editor (TE) was not able to flag these basic axis-label issues within an initial TE review.**
**Overall this is a solid paper, and the article is certainly within scope for ACP, and the findings presented are important, and will be of substantial interest to the community, once the revised manuscript can adequately represent the stratosphere's variation through this period.**
**The scientific writing is generally quite good, and although the writing style is quite brief, the main results are presented in a fair way, aside from the initial Figures 1 and 3.**

**3.1   Main minor revisions**

**M-MR-1) Re-plot Figure 1 with solid black line for GloSSAC (benchmark for stratospheric AOD) As noted in the general comments above, please switch-round the dotted and solid lines given currently for the stratospheric AOD for GloSSAC and OSIRIS.**
**This is needed because the OSIRIS is a limb-scatter instrument, then much less certain aerosol retrieval than for the SAGE-III instrument, providing the main basis of GloSSAC.**
This has been done, GloSSAC (SAGE III, cloud cleared) is now additionally

shown in a thicker line (compared to the figure in the short public reply).

**The colour black should also be reserved for observations, and please also use then change the model lines to be coloured green, with black used for the observation datasets.**
This has been changed for all figures.

**Please also update the y-axis range, to include the peak post-Hunga GloSSAC sAOD values.**
**The GloSSAC observed stratospheric AOD "off the scale" during 2022, in the lower-panel of the current version of Figure 1 (for southern hemisphere stratospheric AOD).**
For easier reading, we use now a linear axis, accommodating even the not cloud cleared SAGE III data of GloSSAC (here dash-dotted).

**M-MR-2) The current Figure 3 needs to be re-drawn for vertically-integrated SAD, to include the main Hunga impacts being at higher altitude than those from the 2019/2020 wildfires.**
**This is particularly evident within Figure 3, the Surface Area Density shown only for 1 selected altitude (68hPa). It is clear when referring to Figures 18 and 19 from Knepp et al. (2024) or Figure 7a from Joerimann et al. (2025) that the observed post-Hunga SAD is significantly higher in magnitude than the wildfires (and also at higher altitude).**
**Had a 50hPa or 30hPa level been chosen for this southern mid-latitude comparison (the upper panel of Figure 3) the reader would have been presented with a timeline showing only a very minor enhancement in 2020 and 2021, and a much larger enhancement in 2022 (see Knepp et al., 2024, Figure 19).**
Our SAD is the one seen by heterogeneous chemistry which differs from the quantities used for calculating extinction. Here a vertical average or integral would be totally misleading because of the strong dependencies on temperature and composition (see above, Knepp et al. (2024) is cited in connection with SAGEIII-ISS in section 2 of the revised manuscript).

**3.2 Other minor revisions**

**O-MR-1) Abstract, line 1 – It's important to be clear the causality requires a conditional clause. Please insert "can" before "affect stratospheric chemistry" (it's only the exceptionally intense wildfires in 2019/2020 that have been shown to have this effect).**
"suggest' implies that already, unfortunately there is a limit in the word count.

**O-MR-2) Abstract, line 3 – Similarly to above, the Abstract needs to be clear it is these particular wildfires that are shown to have**

this effect (the majority of wildfires do not have this effect). Change "emitted from wildfires" to "emitted from intense wildfires in 2019/2020". Already in Fromm et al 2004 and 2005!

**O-MR-3) Abstract, line 4 – Change "due to increased solubility of HCl in particles containing organic acids" – the "due to" is not quite right there – it's the abundance of the smoke in the stratosphere that causes the effect (from the intense fires). Suggested re-wording is "related to a high solubility of HCl in aerosol particles containing organic acids".**
We rephrased the sentence.

**O-MR-4) Abstract, line 5 – Change "the upward transport of the pollution plumes resulted in". The "upward transport" could be misunderstood by some readers to mean a longer timescale effect. But it was actually the extreme intensity of the fire, that then caused a very deep plume, to then detrain the smoke at altitudes above the tropopause. Suggested re-wording is to change "Following the 2019/2020 Australian megfires, the upward transport of the pollution plumes resulted in ozone depletion in the..." to "The extreme intensity of the 2019/2020 Australian wildfires, meant the smoke plume reached the stratosphere, causing in significant ozone depletion in the..."**
Sorry, I have to find a shorter compromise.

**-MR-5) Abstract, line 6 – Please change "It diminished column ozone" – the preceding sentence already stated there was significant ozone depletion, and this "It diminished" phrasing also needs to be improved for peer-reviewed article. Suggest to change "It diminished column ozone in the following two years..." to "With the long residence time in the stratosphere, the wildfire smoke diminished column ozone for the following two years....**
Sorry, too long. We cannot put every details into a word count limited abstract. This should be known by the readers.

**O-MR-6) Introduction, line 16 – Re-word "Since about 2017 it was observed...". This is the 1st sentence of the Introduction, and the "about" is really quite poor wording there. Suggest to adapt the sentence, and cite Mike Fromm's earlier papers (e.g. Fromm et al., 2004, 2005) which pointed out these effects from intense boreal forest fires.**
Adopted.

**O-MR-7) Introduction, lines 19-20 – "This holds for the Australian wildfires" – not clear what was meant here.**
See answer to reviewer 1.

**O-MR-8) Introduction, line 24 – Re-word sentence – improve "like for example Hunga Tonga" for scientific language.**
?? There are more volcanoes in the period which have an impact on the stratosphere but Hunga has the largest.

**O-MR-9) Introduction, line 25 – Similarly, the tense here is not quite right, change "lifting" to "lifted".**
"lifting" was related to "causes".

**O-MR-10) Introduction, line 28 – Again, improve the scientific writing here, re-word "This and the about 10km thick layer" to be more precise. This is also the first time the depth of the smoke aerosol is mentioned, and suggest to hint at the timescale being significant prior to becoming this 10km deep, e.g. maybe "The initial layer of smoke aerosol progressed to a 10km depth enhancement . . . .." or similar.**
Observations indicate 10 to 15 km after 3 weeks. Reference added.

**O-MR-11) Introduction, line 30 – Scientific writing poor here also – "has been analyzed in several studies" – please be clear what type of studies these are – presumably it was meant the observational studies in the initial period after the eruption? If you meant modelling studies, be clear if you mean interactive stratospheric aerosol studies – and there's actually been relatively few of these – the study of Zhu et al. (2022) and Li et al. (2024)**
Several observation and model references included now here.

**O-MR-12) Line 40, heading for section 2 – the "and used satellite data" is not great. And suggest to include the words "interactive aerosol" to be clear these are interactive stratospheric aerosol CCM simulations. Suggest to change to: "Interactive chemistry-climate model simulations and satellite measurement datasets"**
This is in the following text (line 49f). *Optimize title? It might be too long, same holds for OM-MR-16 to 18*

**O-MR-13) Section 2.1 – Line 42 – It seems too abrupt to simply begin "EMAC consists of" – and I think the acronym has not yet been introduced. Suggest to re-word to: "For this study, we analyse interactive stratospheric aerosol simulations with the ECHAM/MESSY Atmospheric Chemistry model (EMAC). . . ".**
Would be a very long sentence. Definition of EMAC already in line 36 as requested by reviewer 1.

**O-MR-14) Line 44 – put "up to 100 hPa" in brackets here, i.e. "(up to 100 hPa)".**

O.K.

**O-MR-15) Line 54 – Re-word "accelerating the classical PCS reactions for chlorine activation also in mid-latitudes". I think you meant that the presence of the extra aerosol means there's substantial additional heterogeneous chemistry occurring in mid-latitude, adding to that on the PSCs at high latitudes. Please re-word to better explain what you meant here.**
I refer now to Table 1 here.

**O-MR-16) Line 80 – Section 2.2 heading – change "OSIRIS" to "OSIRIS aerosol extinction".**
Optional.

**O-MR-17) Line 89 – Section 2.3 heading – change "OMPS-LP" to "OMPS-LP aerosol extinction".**
Optional.

**O-MR-18) Line 99 – Section 2.4 heading: change "AURA-MLS" to "Aura-MLS trace gas retrievals"**
Optional.

**O-MR-19) Lines 105-107 – Please specify which species you are using to evaluate the model here.**
Done.

**O-MR-20) Caption to Figures 1, 2 and 3– The captions to these Figures notes 500 kt SO2 in the red-line sensitivity simulation, but this is difficult for a time-pressed reader to interpret. Please add "All simulations other than the no-Hunga or 500 kt Hunga run apply 400 kt of SO2.**
This is solved by a table requested by reviewer 1 containing the simulations and its line colors.

**References**
Fromm et al. (2004) "New directions: Eruptive transport to the stratosphere: Add fire-convection to volcanoes", Atmos. Environ., vol. 38, 163-165.
Fromm et al. (2005) "Pyro-cumulus injection of smoke to the stratosphere: Observations and impact of a super blow-up in northwestern Canada on 3-4 August 1998", J. Geophys. Res., 110, https://doi.org/10.1029/2004JD005350
Joerimann, A., Sukhodolov, T., Luo, B. et al. (2025), "A REtrieval Method for optical and physical Aerosol Properties in the stratosphere (REMAPv1)", in review, in Geosci. Mod. Dev., https://doi.org/10.5194/egusphere-2025-145
Khaykin, S., Podklajen, A., Ploeger, F. et al. (2022), "Global perturbation of stratospheric water and aerosol burden by Hunga eruption" ,

Comm. Earth Env., https://doi.org/10.1038/s43247-022-00652-x

Knepp, T., Kovilakam, M., Thomason, L. et al. (2024), "Characterization of stratospheric particle size distribution uncertainties using SAGE II and SAGE III/ISS extinction spectra", Atmos. Meas. Tech., 17, 2025–2054, 2024, https://doi.org/10.5194/amt-17-2025-2024

Li, C., Peng, Y, E. Asher et al. (2024) "Microphysical simulations of the 2022 Hunga volcano eruption using a sectional aerosol model", Geophys. Res. Lett., vol. 51, issue 11., https://doi.org/10.1029/2024GL108522

Zhu, Y., Bardeen, C., Tilmes, S. et al. (2022) "Perturbations in stratospheric aerosol evolution due to the water-rich plume of the 2022 Hunga-Tonga eruption", Comms. Earth Env., 3, 248 https://doi.org/10.1038/s43247-022-00580-w

**References**

Ansmann, A., Ohneiser, K., Chudnovsky, A., Knopf, D. A., Eloranta, E. W., Villanueva, D., Seifert, P., Radenz, M., Barja, B., Zamorano, F., Jimenez, C. Engelmann, R., Baars, H., Griesche, H., Hofer, J., Althausen, D., and Wandinger, U.: Ozone depletion in the Arctic and Antarctic stratosphere induced by wildfire smoke, Atmos. Chem. Phys., 22, 11 701–11 726, https://doi.org/10.5194/acp-22-11701-2022, 2022.

Giorgetta, M. A. and Bengsston, L.: Potential role of the quasi-biennial oscillation in the stratosphere-troposphere exchange as found in water vapor in general circulation model experiments, J. Geophys. Res. -Atmos, 104, 6003–6019, 1999.

Giorgetta, M. A., Manzini, E., Roeckner, E., Esch, M., and Bengtsson, L.: Climatology and forcing of the quasi-biennial oscillation in the MAECHAM5 model, J. Climate, 19, 3882–3901, https://doi.org/10.1175/JCLI3830.1, 2006.

Grooß, J.-U., Brautzsch, K., Pommrich, R., Solomon, S., and Müller, R.: Stratospheric ozone chemistry in the Antarctic: what determines the lowest ozone values reached and their recovery?, Atmos. Chem. Phys., https://doi.org/10.5194/acp-11-12217-2011, 2011.

Jöckel, P., Kerkweg, A., Pozzer, A., Sander, R., Tost, H., Riede, H., Baumgaertner, A., Gromov, S., and Kern, B.: Development cycle 2 of the Modular Earth Submodel System (MESSy2), Geosci. Model Dev., 3, 717–752, https://doi.org/10.5194/gmd-3-717-2010, 2010.

Knepp, T., Kovilakam, M., Thomason, L., and Miller, S. J.: Characterization of stratospheric particle size distribution uncertainties using SAGE II and SAGE III/ISS extinction spectra, Atmos. Meas. Tech., 17, 2025–2054, https://doi.org/10.5194/amt-17-2025-2024, 2024.

Santee, M. L., Manney, G. L., Lambert, A., Millán, L. F., Livesey, N. J., Pitts, M. C., Froidevaux, L., Read, W. G., and Fuller, R. A.: The Influence of

Stratospheric Hydration From the Hunga Eruption on Chemical Processing in the 2023 Antarctic Vortex, J. Geophys. Res. -Atmos., 129, https://doi.org/10.1029/2023JD040687, 2024.

Solomon, S., Stone, K., Yu, P., Murphy, D. M., Kinnison, D., Ravishankara, A. R., and Wang, P.: Chlorine activation and enhanced ozone depletion induced by wildfire aerosol, Nature, 615, 259–264, https://doi.org/10.1038/s41586-022-05683-0, 2023.

Stone, K., Solomon, S., Yu, P., Murphy, D. M., Kinnison, D., and Guan, J.: Two-years of stratospheric chemistry perturbations from the 2019/2020 Australian wildfire smoke, Atmos. Chem. Phys., 25, 7683–7697, https://doi.org/10.5194/acp-25-7683-2025, 2025.

[Figure]

Figure 1: Stratospheric aerosol optical depth in tropical latitudes (b), northern (a) and southern midlatitudes (c), zonal averages. Green: EMAC without fires, blue: EMAC with 3 major fires (w/o heterogeneous chemistry on organic aerosol), red: EMAC with heterogeneous chemistry on smoke particles, red dashed: same with 500 kt of $SO_2$ from Hunga Tonga, lightblue: same but $SO_2$ almost co-injected with $H_2O$ but $H_2O$ injection shifted 1.5km upward in a larger box, lightblue dashed: same without $H_2O$ injection, purple: EMAC without Hunga Tonga Black thick: GloSSAC V2.22, observed by SAGE III ISS, cloud cleared, black dash-dotted: GloSSAC V2.22, SAGE III ISS, all, black dotted: observed by OMPS-LP, black dashed: observed by OSIRIS. GloSSAC monthly averages, all other curves 5-day averages.

---

## Author Response (AR2)

**Detailed reply to the referee remarks on "Radiative forcing and stratospheric ozone changes due to major forest fires and recent volcanic eruptions including Hunga Tonga" (second round)**

In the following reviewer remarks are in bold and answers in normal font.

**I would like to thank the authors for addressing my previous comments. I think the paper has made improvements in readability and understanding of the science being done. I appreciate the inclusion of a Table describing the simulations which increases understanding significantly. A few major comments remain below that I feel where not adequately addressed in my previous review. I have added further specific comments that also need to be addressed before publication.**

**Major comments**
**Please include legends in all line figures following ACP author guidelines. It is cumbersome to have to go back to Figure 4 or the Table to see what line is for each Figure. Axis labels should also be included. The figures are still hard to read without this information readily available for each figure.**
Legends and additional short labels are now included in Figs. 1-7 and A1-A5 and the captions shortened.

**Regarding my previous comment on HCl, the author's response states: "The time of the HCl recovery depends on size and depth of the ozone hole (Grooß et al., 2011). If local ozone is close to zero HCl recovers quickly, if some is left, e.g. due to advection, recovery takes longer (see Fig. 7 of preprint). Model uncertainties on this are largest near the vortex edge. Text modified." This does not explain why your simulation disagrees with observations. At 72S, in both your control run and fire runs, HCl recovers a month later than MLS from 2020-2022. Therefore, in your simulations, you have enhanced ClO occurring during a time when HCl in MLS is almost completely recovered. If HCl in your simulations recovered at the same time as MLS, would you still expect to see significant ozone depletion in September all the way to the pole, as your simulations suggest in Figure 8? That seems unlikely to me and therefore your ozone loss results may be overestimated. Please check.**
Text expanded. In EMAC due to some ozone left from numerical diffusion chlorine deactivation occurs also via $ClONO_2$ formation, delaying the increase of HCl. The delay in HCl recovery of model compared to MLS is also present in Fig. 4 of Solomon et al. (2023).

**Specific Comments**
**Line 10. "The sunlight-absorbing aerosol from the Australian and**

Canadian forest fire emissions in 2019/2020 induced the most significant perturbation in stratospheric optical depth since the major eruption of Pinatubo in 1991. It shifted the sign of instantaneous stratospheric aerosol forcing, derived at the top of the atmosphere, from -0.2 Wm-2 to +0.3 Wm-2 in January 2020." Did the Canadian fires really have that large of an effect here? It seems largely insignificant compared to the Australian fires? Please reword.

"to a smaller extent the" inserted before "Canadian". 2017 is now included in most of the figures, more results on the Canadian fires are included in section 3 and the conclusions.

**Line 25 "Because of the absorbing aerosol the radiative impact of forest fire plumes differs from the one of major volcanic eruptions like for example Hunga Tonga in January 2022" Hunga was not like a typical major volcanic eruption though?**

Added at end of sentence "or Raikoke in 2019 where scattering aerosol dominates"

**Line 33: "Hunga Tonga, short Hunga (Santee et al., 2024), the most explosive eruption in the last 3 decades,". This is a little disjointed. Please reword.**

Modified to "Hunga Tonga (now shortly referred to as Hunga (Santee et al., 2024)) ..."

**Line 125: "Considering the organic aerosol from major forest fires is essential for agreement with the observations by OSIRIS and OMPS-LP" Suggest change "considering" to "including".**

O.K.

**Line 127: "0.0005 as can be seen by the black," Can this really be seen in the black curve? Or just the model simulations?**

Sentence rearranged for clarification.

**Line 130-132: "In the tropics the eruptions of Ulawun in June and August 2019 and of St. Vincent in April 2021 have also a large impact on SAOD while the effect of Australian fires with up to 0.0013 is relatively small." Please fix grammar in this sentence. Something like: "In the tropics, the eruptions of Ulawun in June and August 2019 and of St. Vincent in April 2021 also have a large impact on SAOD, while the effect of the Australian fires is relatively small, only up to 0.0013." You mention the ANY enhancement of .0013, but not the Volcanoes even though they are the major cause of the increase.**

Changed as suggested, typical values inserted.

**Line 135-139: "During northern summer 2020 OSIRIS data were so sparse that OMPS-LP had to be used to estimate SO2 injections of some volcanic events and it looks like that the used factors "f" (Schallock et al., 2023) for these were too large, leading to an overestimate of the sulfate contribution to extinction still visible to the end of 2020 in the southern hemisphere" I think you mean: "it looks like they used factors "f"". Please don't speculate here (although I understand this information may not be directly available).**

Modified, now "correction factors "f" by Schallock et al. (2023)". Is defined in reference.

**Line 138-139: "In April 2023 there appears to be a southern midlatitude volcanic event missing in our inventory" I asked this in the previous review. What eruption is missing? Are you referring to the dotted black OMPS-LP line? There seems to be a peak in austral winter in this dataset in the southern midlatitudes for 2019, 2021, and 2023 that is not seen in your model runs or the GLOSSAC dataset? Please check and describe the discrepancies here.**

Unfortunately, I could not identify the possibly missing eruption in the Smithsonian database or on the NASA SO$_2$ website (links in Schallock et al. (2023)). Problem more likely due to model or instrument artifact. Text expanded.

**Line 163: Change "causes also" to "also causes".**

Corrected.

**Line 164-165: Change: "The cooling rate by the injected water vapour we cannot directly derive with our online method." to "The cooling rate by the injected water vapour cannot be directly derived using our online method" or something similar.**

Corrected.

**Line 190-191: "Is strongly enhanced in midlatitudes by up to 0.4 ppmv in agreement with Solomon et al. (2023) and HOCl in high latitudes" This sentence is confusing. I assume you mean HOCl is enhanced at high latitudes. It also looks like it is enhanced at 52S too. It is just hard to see because it is on the same axis as ClONO2.**

Modified. MLS ClO is now also included in Fig.4 for support.

**Line 194-195: "Here we obtain similar results as Zhang et al. (2024b), we found however from a sensitivity study that 195 the reaction HOBr+HCl contributes only a small fraction to chlorine activation." Do you show this anywhere? What was the sensitivity study? Zhang et al looked exclusively in the middle latitudes.**

This refers to a special version of scenario 6 using the * case in Table 1. Text improved, also in the caption of Table 1.

**Line 203-204. "the midlatitude lower stratosphere and almost complete loss at the edge of the ozone hole in winter 2020 in agreement with MLS." Considering the initial offsets between your simulations and MLS in June 2020, it looks like you are overdoing the total depletion compared to MLS, you just arrive at the same minimum.**

A bias (offset) is also in Solomon et al. (2023). An exact agreement cannot expected because of resolution differences and necessary parameterizations with uncertainties.

**Line 205-206: "Simulated ozone changes are consistent with Solomon et al. (2023), Stone et al. (2025) and Zhang et al. (2024a)" They are consistent with Solomon and Stone in the midlatitudes, but not in the polar region.**

Our results look rather similar to Figs. 4 in Solomon et al. (2023) and Stone et al. (2025), and for Hunga, Zhang et al. (2024).

**Line 225: "the remaining effect at TOA was about 0.05 Wm-2" This is .07 on line 152.**

Thanks, corrected.

**Line 228: "Boreal fires in 2017 and 2019 reduced the volcanic forcing at TOA but enhanced it at the tropopause. It is needed to explain the observed AOD" Are the 2017 fires needed? No results are shown before 2019. What volcanic forcing are you referring to? Hunga? Is it the same for the ANY fires? Line 228. What volcanoes are you referring to in 2017 and 2019? Where do you show that the fires reduce the volcanic TOA forcing?**

The figures in the main text include now also 2017 and 2018. The forcing changes by the Boreal fires are included now in the text of section 3.1 and the conclusions. The volcanoes Raikoke and Ulawun in 2019 were mentioned in Section 3.1.

**References**

Santee, M. L., Manney, G. L., Lambert, A., Millán, L. F., Livesey, N. J., Pitts, M. C., Froidevaux, L., Read, W. G., and Fuller, R. A.: The Influence of Stratospheric Hydration From the Hunga Eruption on Chemical Processing in the 2023 Antarctic Vortex, J. Geophys. Res. -Atmos., 129, https://doi.org/10.1029/2023JD040687, 2024.

Schallock, J., Brühl, C., Bingen, C., Höpfner, M., Rieger, L., and Lelieveld, J.: Reconstructing volcanic radiative forcing since 1990, using a comprehensive emission inventory and spatially resolved sulfur injections from satellite data in a chemistry-climate model, Atmos. Chem. Phys., 23, 1169–1207, https://doi.org/10.5194/acp-23-1169-2023, 2023.

Solomon, S., Stone, K., Yu, P., Murphy, D. M., Kinnison, D., Ravishankara, A. R., and Wang, P.: Chlorine activation and enhanced ozone depletion induced by wildfire aerosol, Nature, 615, 259–264, https://doi.org/10.1038/s41586-022-05683-0, 2023.

Stone, K., Solomon, S., Yu, P., Murphy, D. M., Kinnison, D., and Guan, J.: Two-years of stratospheric chemistry perturbations from the 2019/2020 Australian wildfire smoke, Atmos. Chem. Phys., 25, 7683–7697, https://doi.org/10.5194/acp-25-7683-2025, 2025.

Zhang, J., Kinnison, D., Zhu, Y.and Wang, X., Tilmes, S., Dube, K., and Randel, W.: Chemistry Contribution to Stratospheric Ozone Depletion After the Unprecedented Water-Rich Hunga Tonga Eruption, Geophys. Res. Lett., 51, https://doi.org/10.1029/2023GL105762, 2024.

---

## Author Response (AR3)

**Detailed reply to the referee remarks on "Radiative forcing and stratospheric ozone changes due to major forest fires and recent volcanic eruptions including Hunga Tonga" (third round)**

In the following reviewer remarks are in bold and answers in normal font.

**I would like to thank the authors for addressing my comments. I am happy for this paper to be accepted. I have a couple of technical corrections and a minor comment regarding an author response to review listed below.**

**Line 209: "aginst" to "against"**

Corrected.

**Figure 4 panel letters are not aligned and partly hidden.**

Corrected as far as possible in the graphics software.

**Please add y-axis labels to Appendix figures A8 and A9.**

Done.

**In the author's response letter they state: "A bias (offset) is also in Solomon et al. (2023). An exact agreement cannot expected because of resolution differences and necessary parameterizations with uncertainties.". I understand you can't simulate observation concentrations exactly. The point of the comment (and the original comment in the first review) was that your control run is in good agreement with MLS if you take into account your model bias. It seems this is also true for most years shown from 2017-2023, although I am eyeballing. This relates to the sentence on line 214: "almost complete loss at the edge of the ozone hole in winter 2020 in agreement with MLS (black and red curves), in contrast to the control run (green)". You are comparing MLS minimum values to the control run when the control run never captures the MLS seen minima even in years without fires or Hunga? Your statement might be true if you take into account the bias, but at the least, the wording here needs to be changed to acknowledge that your control run minimums don't agree with MLS in any year and therefore your fire simulations that do agree with the minimums are maybe overdoing the total amount of depletion (at least at 68 hPa, it might be different at other levels and in Figure 8).**

We adjusted the respective paragraph as follows:

"Simulated and observed ozone is shown in Fig. 7. The high bias in EMAC lower stratospheric ozone is most likely caused by numerical diffusion and to a small fraction by the underestimated water vapour (Fig. A1) related to a cold bias at the tropical tropopause. In years with weak vortices, numerical diffusion can cause an underestimate of ozone depletion, which is not the case in years with strong vortices (e. g. 2018). Nevertheless, relative effects from fire and volcano emissions can be studied. Here the combined effect of perturbation of

dynamics and chemistry by smoke causes a zonal average decrease of almost 10% in the midlatitude lower stratosphere, almost complete loss at the edge of the ozone hole and complete loss south of 80$^o$S (not shown) in winter 2020 and 2021 in agreement with MLS (black and red curves). In contrast, the remaining ozone in the control run (green) and the run without heterogeneous chemistry on organics (blue) is 0.2 ppmv (80$^o$S) to 0.5 ppmv (72$^o$S) higher. In 2020 the calculated ClO increase near the edge of the Antarctic vortex occurs earlier than in the observations (Fig. 4d) leading to the earlier ozone decrease in Austral fall which reduces the bias between 60 and 80$^o$ S. The effect of Hunga is mostly visible at the vortex edge in October 2022 (b, purple and blue or lightblue curves). Simulated ozone changes are consistent with Solomon et al. (2023), Stone et al. (2025) for the wildfires and Zhang et al. (2024) for Hunga."

The complete ozone loss in the vortex at 80$^o$S is depicted in Fig. 1, which is reproduced by the simulation including wildfires, compared to the control run (green) and the simulation without heterogeneous chemistry on organics (blue).

[Figure]

Figure 1: Simulated and observed O$_3$ at 68 hPa at 60$^o$S (a) and 80$^o$S (b). Legend as in Fig. 7 of the manuscript.

Additionally, we show in Fig. 2 that the model reproduces middle and upper stratospheric and lower mesospheric ozone distributions with latitude

and pressure as observed by MLS, including seasonal effects. Similar figures we have for the other MLS species HCl, ClO, HNO$_3$ and H$_2$O mentioned in the manuscript. MLS HCl appears to be high by at least 5% compared to the observed organic chlorine at ground (AGAGE, NOAA).

**References**

Solomon, S., Stone, K., Yu, P., Murphy, D. M., Kinnison, D., Ravishankara, A. R., and Wang, P.: Chlorine activation and enhanced ozone depletion induced by wildfire aerosol, Nature, 615, 259–264, https://doi.org/10.1038/s41586-022-05683-0, 2023.

Stone, K., Solomon, S., Yu, P., Murphy, D. M., Kinnison, D., and Guan, J.: Two-years of stratospheric chemistry perturbations from the 2019/2020 Australian wildfire smoke, Atmos. Chem. Phys., 25, 7683–7697, https://doi.org/10.5194/acp-25-7683-2025, 2025.

Zhang, J., Kinnison, D., Zhu, Y.and Wang, X., Tilmes, S., Dube, K., and Randel, W.: Chemistry Contribution to Stratospheric Ozone Depletion After the Unprecedented Water-Rich Hunga Tonga Eruption, Geophys. Res. Lett., 51, https://doi.org/10.1029/2023GL105762, 2024.

[Figure]

Figure 2: Observed and calculated ozone (5 day averages) at Jul. 1 and Nov. 1 2020.